# Behavioral role of PACAP signaling reflects its selective distribution in glutamatergic and GABAergic neuronal subpopulations

Limei Zhang[1,2†]*, Vito S Hernandez[1†], Charles R Gerfen[3], Sunny Z Jiang[2], Lilian Zavala[1], Rafael A Barrio[2,4], Lee E Eiden[2]*

[1]Department of Physiology, Faculty of Medicine, National Autonomous University of Mexico, Mexico City, Mexico; [2]Section on Molecular Neuroscience, National Institute of Mental Health, Intramural Research Program, Bethesda, United States; [3]Laboratory of Systems Neuroscience, National Institute of Mental Health, Intramural Research Program, Bethesda, United States; [4]Department of Complex Systems, Institute of Physics, National Autonomous University of Mexico (UNAM), Mexico, Mexico

**Abstract** The neuropeptide PACAP, acting as a co-transmitter, increases neuronal excitability, which may enhance anxiety and arousal associated with threat conveyed by multiple sensory modalities. The distribution of neurons expressing PACAP and its receptor, PAC1, throughout the mouse nervous system was determined, in register with expression of glutamatergic and GABAergic neuronal markers, to develop a coherent chemoanatomical picture of PACAP role in brain motor responses to sensory input. A circuit role for PACAP was tested by observing *Fos* activation of brain neurons after olfactory threat cue in wild-type and PACAP knockout mice. Neuronal activation and behavioral response, were blunted in PACAP knock-out mice, accompanied by sharply downregulated vesicular transporter expression in both GABAergic and glutamatergic neurons expressing PACAP and its receptor. This report signals a new perspective on the role of neuropeptide signaling in supporting excitatory and inhibitory neurotransmission in the nervous system within functionally coherent polysynaptic circuits.

*For correspondence:
limei@unam.mx (LZ);
eidenl@nih.gov (LEE)

[†]These authors contributed equally to this work

Competing interests: The authors declare that no competing interests exist.

## Introduction

Pituitary adenylate cyclase-activating peptide (PACAP) was first isolated from ovine hypothalamic tissue and characterized as a peptide which stimulates cyclic AMP elevation in rat anterior pituitary cells in culture (*Miyata et al., 1989*). PACAP binding to its receptors Vipr1, Vipr2 and, predominantly, PAC1 initiates cell signaling through multiple intracellular pathways (*Harmar, 2001*). PACAP acting at PAC1 is generally considered to engage Gαs, activating adenylate cyclase, with some isoforms also activating phospholipase C via Gαq, leading to multiple cellular responses including increased neuronal excitability (*Pisegna and Wank, 1993*; *Spengler et al., 1993*; *Kawasaki et al., 1998*; *Emery et al., 2013*; *Jiang et al., 2017*; *Johnson et al., 2019*). The PACAP/PAC1 signaling pathway has consistently been related to psychogenic stress responding, and potentiation of this pathway has been linked to psychopathologies including anxiety and PTSD in human (*Ressler et al., 2011*; *Wang et al., 2013a*; *Mustafa et al., 2015*). PACAP gene knock-out in the mouse results in decreased hypothalamo-pituitary-adrenal (HPA) axis activation after physical or psychogenic stress (*Stroth and Eiden, 2010*; *Tsukiyama et al., 2011*), and a hypoarousal behavioral phenotype in response to psychogenic stress (*Lehmann et al., 2013*; *Mustafa et al., 2015*; *Jiang and Eiden,*

*2016*). However, interactions within and among populations of PACAP- and PAC1-expressing neurons in brain circuits mediating behavioral responses to environmental stimulation remain to be understood. This is a critical step in integrative understanding of the functional significance of PACAP-PAC1 neurotransmission.

Exploration of PACAP-containing circuits in rodent CNS has been based on reports of the distribution of PACAP peptide and mRNA, and on expression from reporter genes under the control of a PACAP promoter transgene (*Hannibal, 2002*; *Condro et al., 2016*; *Koves, 2016*) or knocked-in to the PACAP gene itself (*Krashes et al., 2014*). Hannibal reported the anatomical distribution of PACAP projection fields and cell groups in rat CNS employing immunohistochemistry (IHC) and in situ hybridization (ISH), using radiolabeled riboprobes, in a rigorous study. However, due to the paucity of PACAP in cell bodies, dendrites, and axons compared to nerve terminals, peptide IHC has not provided more definitive PACAP chemoanatomical circuit identification in rodent brain. Similarly, ISH with radiolabeled riboprobes, while identifying PACAP-positive cell bodies, lacks the resolution to identify the co-transmitter phenotypes and precise microanatomical features of these cell groups. Thus, heterogeneity of PACAP-containing neurons within and between brain regions, both with respect to cell type and accurate regional boundaries could not be discerned. Nevertheless, an essential function for PACAP as a neurotransmitter within one or more brain behavioral circuits, and consistent with the cellular and post-synaptic actions of PACAP, has not yet emerged. A systematic analysis with accuracy at the level of cellular co-phenotypes, and with anatomical resolution to the level of sub-nuclei within CNS, is essential to complete this task.

To address these issues, we conducted a systematic analysis placing the PACAP>PAC1 signaling into anatomical and basic sensorimotor circuit contexts. We first describe in detail the overall topographical organization of expression of mRNAs encoding PACAP (*Adcyap1*) and its predominant receptor PAC1 (*Adcyap1r1*), and their co-expression with the small-molecule transmitters, glutamate, and GABA, using probes for the expression of mRNAs encoding the vesicular transporters VGLUT1 (*Slc17a7*), VGLUT2 (*Slc17a6*), and VGAT (*Slc32a1*), in mouse brain. We then examined the distribution of PACAP/PAC1 hubs within well-established sensory input-to-motor output pathways passing through the cognitive centers, within the context of glutamate/GABA neurotransmission. This systematic analysis has revealed several possible PACAP-dependent networks involved in sensory integration allowing environmental cues to guide motor output.

## Results

We have studied *Adcyap1* and *Adcyap1r1* co-expression with *Slc17a7*, *Slc17a6*, and *Slc32a1* in the mouse brain, with precise region and subfield identification, using a sensitive dual ISH (DISH) method. *Figure 1*; *Figure 1—source data 1* (*Adcyap1*), *Figure 2* (*Adcyap1r1*) show examples using this method, which unambiguously labels the co-expression of two mRNAs at the single-cell level for *light microscopical* examination. At the light microscopical level, facile low- and high- magnification switching allows detailed serial high-power images to be located in a global histological context for precise delineation of anatomical regions/subfields as well as their rapid photo-documentation. Single-cell co-expression of two mRNA targets can be clearly observed by light microscopy with both low and high magnification.

### Comprehensive DISH mapping of *Adcyap1* co-expression with Slc17a7, Slc17a6, and Slc32a1 throughout mouse brain reveals an extensive distribution and diversity of cell types

*Table 1* describes the distribution, cell types and relative expression strength, within 180 identified *Adcyap1* positive cell groups/subfields co-expressing vesicular transporters, organized hierarchically by grouping the regions according to their embryonic origins. These include 58 regions derived from cortical plate, 6 regions derived from cortical subplate, 9 regions within cerebral nuclei, 20 regions in thalamus, 2 regions in epithalamus, 26 regions in hypothalamus, 18 regions in midbrain, 16 regions in pons, 20 regions in medulla, and 5 regions in cerebellum. We semiquantitative scoring is specified in the Materials and method section. Briefly, our annotation criteria were the percentage of expressing cell/total Nissl-stained nuclei: '-', not observed; '+', weak (<20%); '++', low (20%–40%); '+++', moderate (40%–60%); '++++', intense (60–80%); '+++++', very intense (>80%). Functional neuroanatomy order and annotations are based on Allen Institute Mouse Reference Atlas

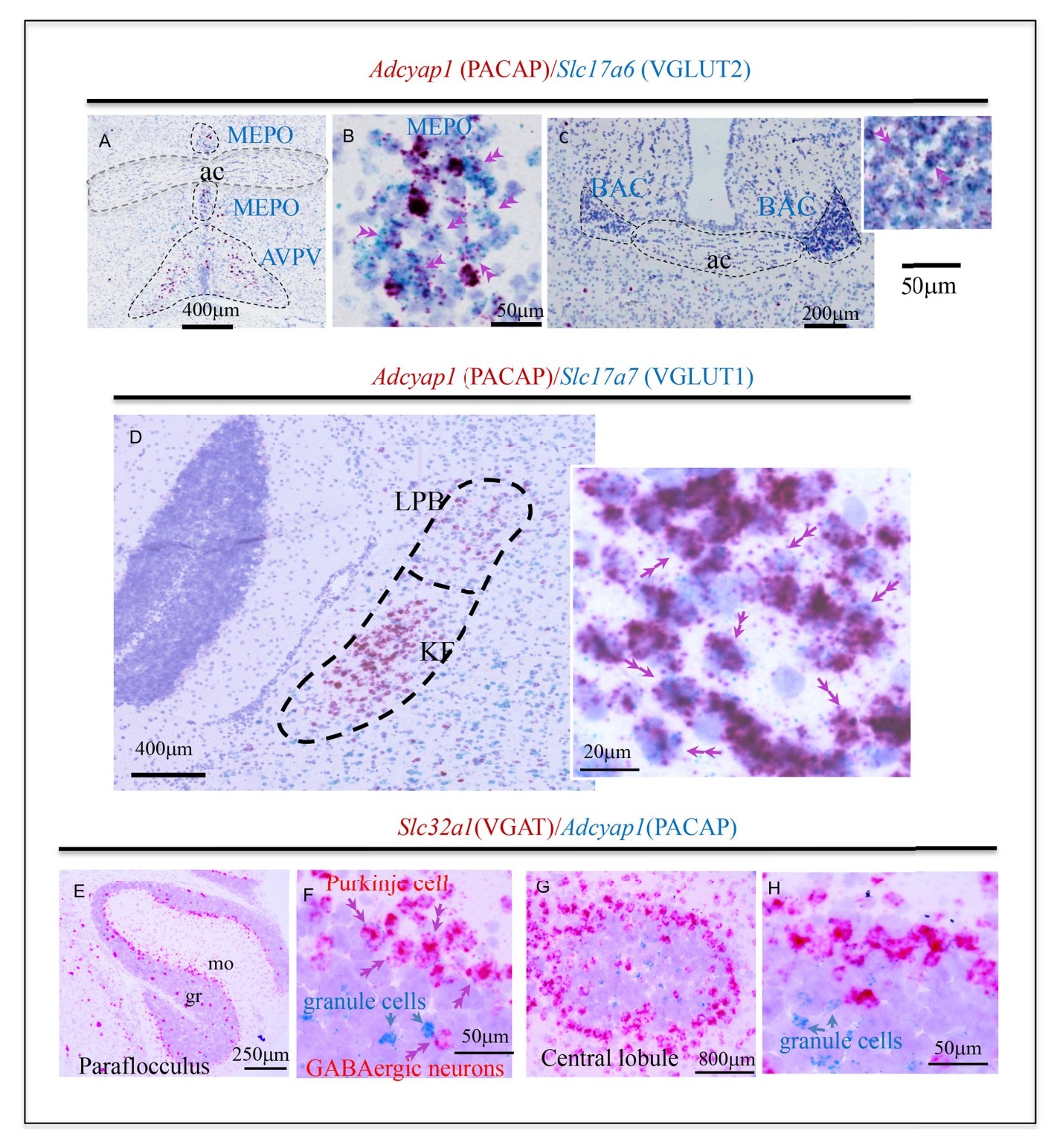

**Figure 1.** Examples of histological samples using the sensitive dual in situ hybridization (DISH) method that can label unambiguously the co-expression of two RNAs at single-cell level for light microscope examination. (**A** and **C**) *Adcyap1* (RNA coding for PACAP) co-expression with *Slc17a6* (RNA coding for VGLUT2) in MEPO and BAC respectively. (**B**) High magnification of the MEPO region of (**A**). Double arrows indicate cells that co-express both mRNAs. This feature is better appreciated on the cells in which *Adcyap1* is weakly expressed (weak staining), so the Slc17a6 staining can be clearly seen as independent dots. (**D**) Brain stem Koelliker Fuse (KF) nucleus of the parabrachial complex is another main PACAP-expressing nucleus. *Adcyap1* was co-expressed intensely with *Slc17a7* (RNA coding for VGLUT1). Double arrows indicate cells co-expressing both mRNAs. Panels E-H show two

*Figure 1 continued on next page*

*Figure 1 continued*

cerebellar regions, paraflocculus (**E and F**) and central (**G and H**) lobules, under low and high magnification, respectively, where the *Adcyap1* expression was higher than rest of regions. Purkinje cells are the main GABAergic (expressing *Slc32a1*, RNA encoding VGAT) PACAP containing neurons, distributed in all regions of cerebellar cortex. Some GABAergic cells in granule cell layer of paraflocculus and central regions also co-expressed *Adcyap1* and *Slc32a1* (indicated with double pink arrowheads, see also SI-Fig.O1). In these two cerebellar regions, some granule cells also expressed *Adcyap1* (indicated with single blue arrows). Nissl staining was used for counterstaining. Note: this figure contains excerpts from the more comprehensive figure supplement.

The online version of this article includes the following source data for figure 1:

**Source data 1.** Comprehensive DISH mapping of PACAP co-expression with VGLUT1, VGLUT2, and VGAT throughout mouse brain reveals an extensive distribution and diversity of cell types.

(http://atlas.brain-map.org/). To compare with the previous comprehensive report for PACAP distribution in rat brain published in 2002 (*Hannibal, 2002*), a column containing the data published previously in rat is displayed. Most of the regions described as *Adcyap1*-expressing in the rat were also found positive in our study in mouse, albeit strength of expression in several regions differs substantially between the two rodent species. Eleven regions that were reported negative, labeled as '-' from original publication, were found positive with this sensitive DISH method (indicated in the table). An additional 122 regions, which were not reported in detail in the previous paper (labeled in the table as 'n/r'), were found to co-express *Adcyap1* and either a glutamate or GABA vesicular transporter mRNA.

A whole brain mapping of *Adcyap1*expression with relevant brain regions/subfields co-expression features is presented in the *Figure 1—source data1*.

## Mapping of *Adcyap1r1* co-expression with *Adcyap1*, *Slc17a7*, Slc17a6, and *Slc32a1* suggests the PACAP-PAC1 system can function in *autocrine* and *paracrine* modes

*Adcyap1r1* expression was studied in 152 mouse brain regions. In *Figure 3*, panels A–F, we show the semi-quantitative expression levels of *Adcyap1r1*, based on microscopic observation as different intensities of blue shading. *Adcyap1* expression was also symbolized with either red or green dots (VGAT vs VGLUT mRNA co-expression) in corresponding regions. Contrasting with the discrete expression of *Adcyap1*, *Adcyap1r1* expression was diffuse and widespread. *Adcyap1r1*-positive cells co-expressed *Slc17a7* in the temporal hippocampus (*Figure 2* panels A and A'), anterior cingulate area (ACA, panel B) and bed nucleus of anterior commissure (BAC, panel C) and *Scl32a1* in pallidum and striatum structures (*Figure 2D–G* and *Table 2*). Almost all the *Adcyap1*-expressing neurons we studied co-expressed *Adcyap1r1* (*Figure 3G*, from ACA and H from medial preoptic nucleus, MEPO). Besides, most neurons neighboring *Adcyap1*-positive cells also expressed *Adcyap1r1* (single arrows). These observations suggest that the PACAP/PAC1 pathway may use *autocrine* and *paracrine* mechanisms in addition to classical neurotransmission through axon innervation and transmitter co-release.

PACAP also binds to two other G protein–coupled receptors highly related to PAC1, called VPAC1 (Vipr1), and VPAC2 (Vipr2) (*Harmar, 2001*). In all the regions where *Adcyap1r1* was expressed, the expression of mRNA for either or both VIP receptors (*Vipr1* and *Vipr2*) was also found. To simplify this already extensive report, we present the data for these two receptors in *Figure 3—source data 1–4* .

Distribution and glutamatergic/GABAergic vesicular transporter mRNA co-expression with Adcyap1 and Adcyap1r1 suggests a broad function for PACAP signaling in sensorimotor processing system(s)

### Retina

Retinal ganglion cells (RGC) have been reported to express PACAP at various levels of abundance previously in the literature. In rat, *Adcyap1* was reported at a *low* level ('+') (*Hannibal, 2002*) within the RGC population. CD1 mice were reported to express PACAP in retina (*Kawaguchi et al., 2010*), and in *Adcyap1* promoter-EGFP reporter mice, EGFP expression was reported to be *low* ('+') (*Condro et al., 2016*). With the DISH method employed here, we found a higher percentage of RGCs co-expressing *Adcyap1* and *Slc17a6* than previously reported (*Figure 1—source data 1A1*).

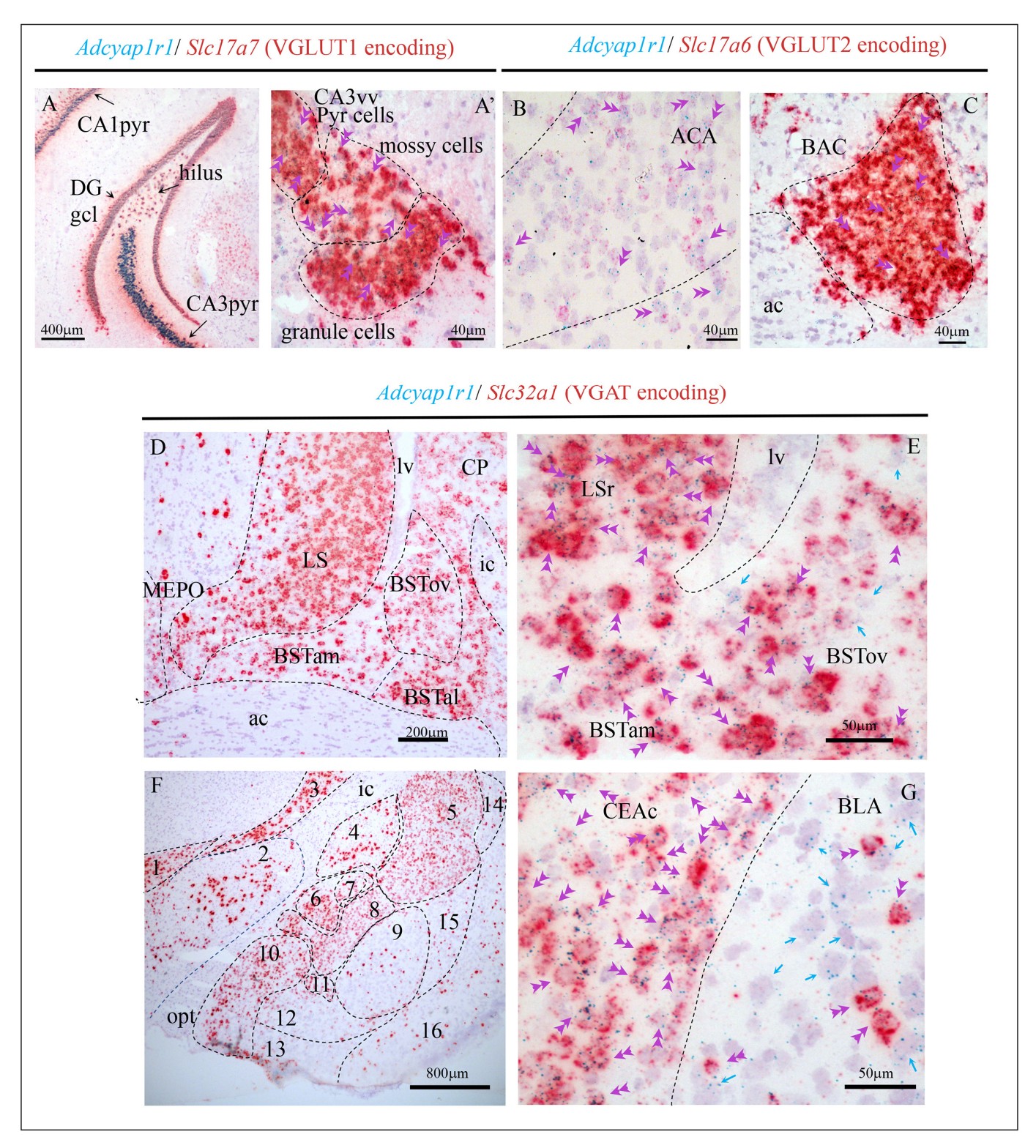

**Figure 2.** Examples illustrating Adcyap1r1 (RNA encoding PAC1) co-expression with glutamatergic (*Slc17a7*-VGLUT1 and *Slc17a6*-VGLUT2 expressing) and GABAergic (*Slc32a1*-expressing) neurons in cortical and subcortical regions. (**A** and **A'**) Temporal hippocampal formation where the *Adcyap1r1* was strongly expressed in the principal neurons (pyr: pyramidal layer and DGgcl: dentate gyrus granule cell layer) as well as the VGLUT1+ mossy cells in the hilar region. Double arrows show single cells co-expressing Adcyap1r1 and *Slc17a7*. (**B**) Single-cell *Adcyap1r1* co-expression with *Slc17a6* (double arrowheads) was observed in ACA and (**C**) in the BAC. Regarding the GABAergic neurons expressing *Adcyap1r1*, the structures in the striatum and

*Figure 2 continued on next page*

*Figure 2 continued*

pallidum hosted very intensely expressing structures. Panel (**D**) shows the LS and BST in its three anterior divisions, anteromedial (BSTam), antero-lateral (BSTal) and oval (BSToval), as well as caudate-putamen (CP) with strong *Adcyap1r1* expression. Panel (**E**) shows high-magnification photomicrograph where green dots (*Adcyap1r1*, PAC1 labeling), are mostly overlapped with red staining (*Slc32a1*, VGAT expression). Double pink arrowheads indicate co-expression within a single cell and single green arrows indicate cells only expressing *Adcyap1r1*. Panel (**F**) shows the amygdaloid complex and neighboring regions where *Adcyap1r1* was strongly expressed in the GABAergic cell populations; (**G**) High-magnification photomicrograph showing that the *Adcyap1r1* is exclusively expressed in *Slc32a1* (VGAT) expressing neurons in the CEAc, while in the BLA it was expressed in the sparsely distributed GABAergic neurons as in most of the non-VGAT expressing neurons. 1. zona incerta of hypothalamus; 2. lateral hypothalamic area; 3: reticular nucleus of the thalamus; 4. globus pallidus; 5. caudate-putamen; 6: central amygdalar nucleus, medial part (CEAm); 7: lateral part (CEAl); 8: capsular part (CEAc); 9: basolateral amygdalar nucleus (BLA) 10: medial amigdalar nucleus; 11: intercalated nucleus of the amygdala; 12: basomedial nucleus of the amygdala; 13: cortical amygdalar area; 14: dorsal endopiriform; 15: ventral endopiriform; 16: piriform area. Fiber tracts: sm: stria medullaris; ac: anterior commissure; ic: internal capsule; opt: optic tract. lv: lateral ventricle.

Expression levels of *Adcyap1* oscillate daily from 50% to 80% with highest levels during subjective night (see *Lindberg et al., 2019* for details).

## Cerebral cortex: structures derived from cortical plate

### Olfactory area

High levels of PACAP expression in the olfactory area have been previously reported (*Hansel et al., 2001*). Here, we report the subfields of olfactory area *Adcyap1*-expressing neurons in detail. In the *main olfactory bulb* (MOB, *Figure 3B*, area 1), *Adcyap1* was *intensely* expressed in outer plexiform (OPL) and mitral layers. In OPL we observed the co-expression with *Slc17a7*, *Slc17a6* and *Slc32a1*. (*Table 1* and *Figure 1—source data 1A2* and insets). Other cell types in the internal plexiform cell layers expressed *Adcyap1* at low levels with mixed glutamate/GABA molecular signatures (see *Table 1*). In contrast, in the *accessory olfactory bulb* (AOB, *Figure 3C*, area 83), *Adcyap1* was mainly expressed in the mitral cell layer with co-expression of *Slc17a7*, *Slc17a6* (*intense*), and *Slc32a1* (*weak*) mRNAs (*Table 1*). Other olfactory areas *intensely* expressing *Adcyap1* and Slc17a7 were AON (layer 1, *Figure 3C*, area 82), TT (*Figure 3C*, area 34b), DPA (*Figure 3C*, area 34a), Pir (layer 3, *Figure 3D and E*, area 129), NLOT (layer, *Figure 3D*, area 108; SI *Figure 1F*). The COA (layer, *Figure 3D*, areas 107a and 107b) co-expressed *Adcyap1 and Slc17a6* (*Figure 1—source data 1F and F5*).

### Isocortex

PACAP's role in isocortex has in general been little studied (*Zhang and Eiden, 2019*). Moderate expression of *Adcyap1* was initially reported in the cingulate and frontal cortices, with lower concentrations found in other neocortical areas using radiolabeled riboprobe ISH (*Mikkelsen et al., 1994*). Hannibal subsequently reported that *Adcyap1*-expressing cells were observed mainly in layers 1–3 and layers 5–6, and PACAP-IR nerve fibers in all layers of the cerebral cortex; however, no detailed information about the differential expression levels across cortical regions was presented (*Hannibal, 2002*).

In our study, we found *intense* expression of *Adcyap1* in isocortex to be in the frontal pole of the telencephalon, including prelimbic, infralimbic and anterior cingulate and orbital area (*Figure 3*, A and B, areas 43, 42, 32a and 32b, and 31, respectively). Approximately 80% of the neuronal population of the layer 2 and layer 5 co-expressed *Adcyap1* and *Slc17a7*. A significant population of *Adcyap1-expressing* cells in the layer 5 of prefrontal cortices co-expressed *Slc17a6* or *Slc32a1* (*Table 1* and *Figure 1—source data 1B and B1*).

In the primary and secondary motor cortices (MOp and MOs, *Figure 3B*, area 44), *Adcyap1* was found expressed in layer 2/3 and layer 5. This pattern was also observed in somatosensory, gustatory, auditory, visual, visceral, temporal association, ectorhinal, perirhinal, retrosplenial, and post-parietal association areas (see *Table 1* for more cortical area expression and strength and *Figure 1—source data 1A–K*, low-magnification panels).

*Adcyap1r1* expression in neocortex was widespread with more homogenous aspects concerning the different cortical areas, except that in the ACA and the entorhinal cortex layers 2–3 and layers 5–6 showed *very intense* expression levels (*Figure 3*, panels B and F, areas 32 and 32a, 139 and 140) and https://gerfenc.biolucida.net/images?selectionType=collection&selectionId=98. We sampled

**Table 1.** Distribution, cell types, and strength of main PACAPergic cell groups in mouse brain with comparison of rat brain reported by *Hannibal, 2002*\*.

| Cell group / sub-field[†] | Hannibal, 2002 | Slc17a7 (VGLUT1) | Slc17a6 (VGLUT2) | Slc32a1 (VGAT) |
|---|---|---|---|---|
| Retina | | | | |
| Ganglion cell layer[‡] | + | - | +++ | - |
| Cerebrum: Cortical plate | | | | |
| Olfactory area | | | | |
| Main olfactory bulb | | | | |
| Granular cell layer | - | - | - | - |
| Inner plexiform layer | - | - | - | - |
| Mitral cell layer | + | +++ | ++ | + |
| Outer plexiform layer | n.r. | +++ | +++ | + |
| Glomerular layer | - | + | + | + |
| Periglomerular cells | n.r. | + | + | + |
| Accessory olfactory bulb | | | | |
| Mitral cell layer | + | ++++ | ++ | + |
| Glomerular layer | n.r. | + | + | + |
| Granular layer | n.r. | + | - | - |
| Other olfactory areas | | | | |
| Ant olfactory n. lateral | ++ | ++++ | ++ | - |
| Ant olfactory n. medial | ++ | ++++ | ++ | - |
| Dorsal peduncular area | n.r. | +++ | ++ | ++ |
| Taenia Tecta | n.r. | +++ | - | - |
| Piriform area: Pir2 | n.r. | + | - | - |
| Piriform area: Pir3 | n.r. | ++++ | + | - |
| N. lat. olfactory tract (NLOT) | ++++ | ++++ | ++ | + |
| Cortical amygdalar area (CoA) | n.r. | + | ++++ | - |
| Hippocampal formation | | | | |
| *Hippocampal region* | | | | |
| Dorsal dentate gyrus | n.r. | - | - | - |
| Dorsal hippocampus CA1 | + | - | - | - |
| Dorsal hippocampus CA2 | n.r. | + | - | - |
| Dorsal hippocampus CA3 | + | - | - | - |
| Dorsal hilus | n.r. | ++ | - | - |
| Ventral dentate gyrus | n.r. | - | - | - |
| Ventral CA3vv | n.r. | +++++ | - | - |
| Ventral hilus | n.r. | ++ | - | - |
| *Retrohippocampal regions* | | | | |
| Entorhinal area | n.r. | + | + | - |
| Parasubiculum | ++ | +++ | +++ | - |
| Postsubiculum | ++ | +++ | +++ | - |
| Presubiculum | n.r. | +++ | +++ | - |
| Subiculum | n.r. | +++ | +++ | - |
| Isocortex[§] | | | | |
| Layer I | + | n.a | n.a | n.a |
| Layer II-II | ++ | n.a | n.a | n.a |

*Table 1 continued on next page*

*Table 1 continued*

| Cell group / sub-field[†] | Hannibal, 2002 | Slc17a7 (VGLUT1) | Slc17a6 (VGLUT2) | Slc32a1 (VGAT) |
|---|---|---|---|---|
| Layer IV | - | n.a | n.a | n.a |
| Layer V | ++ | n.a | n.a | n.a |
| Layer VI | + | n.a | n.a | n.a |
| Agranular insular cortex | n.r. | ++++ | ++ | - |
| Somatomotor areas | | | | |
| 2ry motor area, layer 2–3 | n.r. | +++ | - | - |
| 2ry motor area layer 5 | n.r. | ++++ | ++ | + |
| 1ry motor area, layer 2–3 | n.r. | +++ | - | - |
| 1ry motor area, layer 5 | n.r. | ++++ | ++ | + |
| Orbital frontal cortex (OFC) | | | | |
| OFC 1 | n.r. | ++ | ++ | - |
| OFC 2/3 | n.r. | +++ | + | - |
| OFC 5 | n.r. | +++ | - | - |
| Prefrontal cortex (PFC) | | | | |
| Ant cingulate cortex (ACC): | n.r. | ++++ | + | - |
| ACC 2/3: | n.r. | +++ | + | - |
| ACC 5: | | | | |
| Prelimbic (PL) | n.r. | ++++ | + | - |
| PL 2/3 | n.r. | +++ | + | - |
| PL 5 | | | | |
| Infralimbic (IL) | n.r. | ++++ | + | - |
| IL 2/3 | n.r. | +++ | + | - |
| IL 5 | | | | |
| Cell group / sub-field[†] | Rat Hannibal JCN, 2002 | Slc17a7 (VGLUT1) | Slc17a6 (VGLUT2) | Slc32a1 (VGAT) |
| Prim somatosensory a. SSp, | - | + | - | - |
| SSp 1 | - | +++ | - | - |
| SSp 2/3 | n.r. | ++ | - | - |
| SSp 4 (mouth) | - | ++ | - | - |
| SSp 5 | - | ++ | - | - |
| SSp 6a | | | | |
| Gustatory areas | n.r. | ++++ | - | - |
| Auditory area | n.r. | +++ | - | - |
| Visual area | n.r. | +++ | - | - |
| Visceral area | n.r. | +++ | - | - |
| Temporal association area | n.r. | +++ | - | - |
| Ectorhinal area | n.r. | +++ | - | - |
| Perirhinal area | n.r. | ++ | - | - |
| Retrosplenial area | n.r. | ++++ | - | - |
| Post parietal association area | n.r. | ++++ | - | - |
| Cortical subplate | | | | |
| Claustrum | n.r. | + | + | - |
| Endopiriform nucleus | n.r. | + | + | - |
| Lateral amygdalar nucleus | n.r. | ++++ | ++ | - |
| Post amygdalar nucleus (PA) | n.r. | ++++ | + | - |
| Basomedial amygdala | - | + | + | - |
| Basolateral amygdala | + | + | + | - |
| Cerebral nuclei | | | | |

*Table 1 continued on next page*

*Table 1 continued*

| Cell group / sub-field[†] | Hannibal, 2002 | Slc17a7 (VGLUT1) | Slc17a6 (VGLUT2) | Slc32a1 (VGAT) |
|---|---|---|---|---|
| Striatum | | | | |
| Lateral septal nucleus | n.r. | ++ | ++ | - |
| Anterior amygdala area | n.r. | + | + | - |
| Central amygdalar nucleus | + | - | - | + |
| Medial amygdalar nucleus | ++ | +++ | +++ | + |
| Pallidum | | | | |
| Bed nucleus of Stria Terminalis (BST) | + | n.a | n.a | n.a |
| BST oval | n.r. | - | + | + |
| BST am | n.r. | - | ++ | ++ |
| BST dm | n.r. | - | + | + |
| BST pr | n.r. | + | ++ | ++ |
| Bed nucleus of anterior commissure | n.r. | +++++ | +++++ | - |
| Brain stem, inter-brain | | | | |
| Thalamus | | | | |
| *Somato-motor related* | | | | |
| Subparafacicular nucleus, magnocellular part | n.r. | + | ++ | - |
| Subparafacicular area | n.r. | - | +++ | - |
| Peripeduncular nucleus | n.r. | - | +++ | - |
| Medial geniculate complex | n.r. | - | +++ | - |
| *Polymodal association cortex related* | | | | |
| Lat. Posterior n. thal | n.r. | + | ++ | - |
| Post. Limiting nucleus | n.r. | + | ++ | - |
| Suprageniculate n. | n.r. | + | ++ | - |
| Anterodorsal n. | - | +++ | + | - |
| Anteromedial n. | n.r. | ++ | ++ | - |
| Parataenial n. | n.r. | ++ | ++ | - |
| Intermedial n. | n.r. | + | + | - |
| Laterodorsal n. | n.r. | + | ++ | - |
| Centrolateral n | n.r. | - | ++ | - |
| Intermediodorsal n. | n.r. | + | ++ | - |
| Mediodorsal n. | n.r. | +++ | + | - |
| Pariventricular n. | - | + | +++ | - |
| Parateanial n. | n.r. | + | ++ | - |
| N. of reuniens | - | + | +++ | - |
| Posterior pretectal n. | +++ | - | +++ | - |
| Precommissural n. | +++ | - | + | - |
| Cell group / sub-field[†] | Rat Hannibal JCN, 2002 | Slc17a7 (VGLUT1) | Slc17a6 (VGLUT2) | Slc32a1 (VGAT) |
| Epithalamus | | | | |
| Medial habenula[¶] | ++++ | ++++ | ++++ | - |
| Lateral habenula | ++++ | - | ++++ | - |
| Hypothalamus | | | | |
| Paraventricular n | + | - | + | - |
| Periventricular n | + | - | ++ | - |

*Table 1 continued on next page*

*Table 1 continued*

| Cell group / sub-field[†] | Hannibal, 2002 | Slc17a7 (VGLUT1) | Slc17a6 (VGLUT2) | Slc32a1 (VGAT) |
|---|---|---|---|---|
| Anterodorsal preoptic n. | n.r. | - | + | - |
| Anteroventral | n.r. | - | +++ | - |
| Dorsomedial n. | +++ | - | +++ | - |
| Median preoptic n. | +++ | - | ++++ | - |
| Medial preoptic area | +++ | - | ++ | - |
| Vascular organ of lamina terminalis | +++ | - | ++++ | - |
| Posterodorsal preoptic n. | n.r. | - | + | - |
| Subfornical organ | ++++ | - | ++++ | - |
| Lateral preoptic area | n.r. | - | ++ | - |
| Anterior hyp. area | ++ | - | ++ | - |
| Premammillary n. | n.r. | - | ++ | + |
| Lateral mammillary n. | ++++ | - | ++++ | - |
| Medial mammillary n. | - | - | +++ | - |
| Supramammillary n. | - | - | ++ | + |
| Median preoptic n. | ++ | - | ++ | - |
| Lateral hyp. area | ++ | - | ++ | - |
| Preparasubthalamic n. | n.r. | - | +++ | - |
| Parasubthalamic n. | n.r. | - | +++ | - |
| Subthalamic nucleus | - | - | +++++ | - |
| Retrochiasmatic area | n.r. | - | +++ | - |
| Tuberomammillary nucleus | - | - | ++ | + |
| Zona incerta | + | - | ++ | - |
| Ventromedial hyp. n | ++++ | - | +++++ | - |
| Post. hypothalamic n. | n.r. | - | +++ | - |
| Midbrain | | | | |
| *Sensorial related* | | | | |
| Inf. colliculus (IC), central and external n. | n.r. | - | ++ | - |
| N. of the brachium of IC | n.r. | - | ++ | - |
| N. saculum | n.r. | - | + | - |
| Parabigeminal n. | n.r. | - | + | - |
| Midbrain trigeminal n. | n.r. | - | ++ | - |
| *Motor related* | | | | |
| Ventral tegmental area | n.r. | - | ++ | - |
| Midbrain reticular n. | n.r. | - | + | - |
| Superior colliculus, motor related | n.r. | - | +++ | - |
| Periaqueductal gray | n.r. | - | +++ | - |
| Cuneiform n. | n.r. | - | ++ | - |
| Edinger-Westphal n. | n.r. | - | + | - |
| Interfascicular n. Raphe | n.r. | - | ++ | - |
| *Behavior state related* | | | | |
| Midbrain raphe nuclei | n.r. | - | - | - |
| Pedunculopontine n. | n.r. | - | ++ | - |
| Dorsal n. raphe | n.r. | - | - | - |

*Table 1 continued on next page*

*Table 1 continued*

| Cell group / sub-field[†] | Hannibal, 2002 | Slc17a7 (VGLUT1) | Slc17a6 (VGLUT2) | Slc32a1 (VGAT) |
|---|---|---|---|---|
| Central linear n. raphe | n.r. | - | ++ | - |
| Rostral linear n. raphe | n.r. | - | - | - |
| Olivary pretectal nucleus | n.r. | - | +++ | - |
| Cell group/sub-field[†] | Rat Hannibal JCN, 2002 | Slc17a7 (VGLUT1) | Slc17a6 (VGLUT2) | Slc32a1 (VGAT) |
| Hindbrain | | | | |
| Pons | | | | |
| *Sensory related* | | | | |
| N. lateral lemniscus | n.r. | - | + | - |
| Principal sensorial nucleus of trigeminal nerve | - | - | + | - |
| Koelliker-Fuse subnucleus | n.r. | ++++ | - | - |
| Parabrachial n. lateral div. | n.r. | - | ++++ | - |
| Parabrachial n, rest subfields | n.r. | - | +++ | - |
| Superior olivary comp (lat) | n.r. | + | - | - |
| *Motor related* | | | | |
| Tegmental reticular n. | n.r. | - | +++ | - |
| Barrington's nucleus | n.r. | - | +++ | - |
| Dorsal tegmental n. | n.r. | - | + | - |
| Pontine gray | n.r. | - | +++ | - |
| Pontine central gray | n.r. | - | + | - |
| Supratrigeminal nucleus | n.r. | - | + | - |
| *Behavior state related* | | | | |
| Locus Coerulus (state) | + | ++ | +++ | - |
| Laterodorsal tegmental n. | + | - | +++ | - |
| Pontine reticular n. | n.r. | - | + | - |
| Superior central n. raphe | n.r. | - | + | + |
| Medulla | | | | |
| N. tractus solitarii medial | +++ | ++ | ++++ | - |
| N. tractus solitarii lateral | +++ | - | ++++ | - |
| Hypoglossal (XII) n. | - | | ++ | - |
| Dorsal motor n. of the vagus nerve (X) | +++ | +++ | - | - |
| Dorsal cochlear n. | +++ | ++ | ++ | - |
| Ventral cochlear n. | n.r. | ++ | ++ | - |
| Spinal n. trigeminal | n.r. | - | ++ | - |
| N. prepositus | n.r. | - | ++ | - |
| Inferior salivatory complex | n.r. | - | ++ | - |
| Facial motor n. (VII) | n.r. | - | ++ | - |
| N. ambiguus | +++ | - | ++ | - |
| Magnocellular reticular n. | n.r. | - | ++ | - |
| Parapyramidal n. | n.r. | - | ++ | - |
| Spinal vestibular n. | +++ | - | ++ | - |
| N. X | n.r. | - | + | - |
| N. raphe magnus (state related) | n.r. | - | ++ | - |
| N. raphe pallidus (state related) | n.r. | - | ++ | - |

*Table 1 continued on next page*

*Table 1 continued*

| Cell group / sub-field[†] | Hannibal, 2002 | Slc17a7 (VGLUT1) | Slc17a6 (VGLUT2) | Slc32a1 (VGAT) |
|---|---|---|---|---|
| N. raphe obscurus (state rel.) | n.r. | - | ++ | - |
| Cuneate n. | - | ++ | ++ | - |
| Inferior olivary | n.r. | - | ++ | - |
| Cerebellar cortex | | | | |
| Purkinje's cells | ++ | - | - | +++++ |
| Golgi's cells | n.r. | - | - | + |
| Granule cells[¶] | - | ++ | - | - |
| Cerebellar nuclei | | | | |
| Interposed n. | ++ | + | - | - |
| Dentate n. | n.r. | - | + | - |

n.a.: not applicable.

n.r.: not reported (blue color text refers to **Hannibal, 2002**) rat PACAPergic cell group and expression strength analysis.

*Similar semiquantitative annotations are used here the percentage of expressing cell/total Nissl stained nuclei: '-', not detectable; '+', weak (<20%); '++', low (40–20%); '+++', moderate (60–40%); '++++', intense (80–60%); '+++++', very intense (>80%).

[†] Functional neuroanatomy order and annotations are based on Allen Institute Mouse Reference Atlas.

‡ Circadian oscillating expression (**Lindberg et al., 2019**).

§ Isocortex expression was regionally evaluated.

[¶] Dorsal half of the MHb which co-express *Calb2* (RNA encoding calretinin).

** Prominent in lobules paraflocculus, central and uvula. Coincide with calretinin (*Calb2*) expression.

eight neocortex regions at two coronal levels, Bregma 0.14 mm and Bregma 1.7 mm, where we observed that more than 80% of neurons in layers 2–3 and layer 5 expressed *Adcyap1r1* (*Figure 3—source data 5*). As approximately 20% of cortical neurons are GABAergic (*Ascoli et al., 2008*), we tested three of the main GABAergic cell types in these cortical regions, finding that in the selected cortical areas we sampled, all of somatostatin (Sst), parvalbumin (PV) and corticotropin releasing hormone (CRH) neurons co-expressed *Adcyap1r1* (*Figure 3—figure supplement 1 F,G,H*).

## Hippocampal formation

In the mouse dorsal (septal pole) hippocampal formation, in contrast to data obtained in rat (*Figure 3—figure supplement 2*) and from the PACAP-EGFP transgenic reporter mouse (*Condro et al., 2016*), we did not find *Adcyap1*-expressing cells in cell body layers of CA1, CA3, and DG, as previously reported (*Hannibal, 2002*; *Condro et al., 2016*). However, we report here the marked and selective expression of *Adcyap1* in pyramidal neurons of the CA2 region (*Table 1* and *Figure 4C*, left inset). *Adcyap1r1* expression was observed to be *low* in CA subfields and *Adcyap1r1* was selectively expressed in *Slc32a1*-expressing cells (*Figure 3—figure supplement 1, A and B*). In contrast, the DG-GCL had *very intense* expression level of *Adcyap1r1*, among all brain regions, in both Slc17a7- and *Slc32a1*-expressing cells (*Figure 2A* and *Figure 3—figure supplement 1, A and C*). In DG hilar region (polymorphic layer), we observed few cells co-expressed *Adcyap1* and *Slc17a7* (*Figure 4A and D*). These were *mossy cells* co-expressing calretinin mRNA (*Calb2*) (*Figure 4B*). The *Adcyap1*-expressing mossy cell quantity increased in the caudo-temporal direction. This population was also described in the previous reports (*Hannibal, 2002*; *Condro et al., 2016*), however, without identification of cell type, as reported here.

In the ventral (temporal pole) hippocampal formation, we identified two cell populations with *Adcyap1* expression. One was the *Slc17a7*-expressing mossy cell population in the hilar region mentioned above, which was distributed from septo-dorsal to temporo-ventral hilus with increasing quantity (*Figure 4C* and *Figure 1—source data 1J*) Mossy cells are major local circuit integrators and they exert modulation of the excitability of DG granule cells (*Scharfman and Myers, 2012*; *Sun et al., 2017*). The DG granule cells strongly expressed *Adcyap1r1* (*Figure 3—figure supplement 1A and C*). Glutamatergic hilar mossy cells of the dentate gyrus can either excite or inhibit

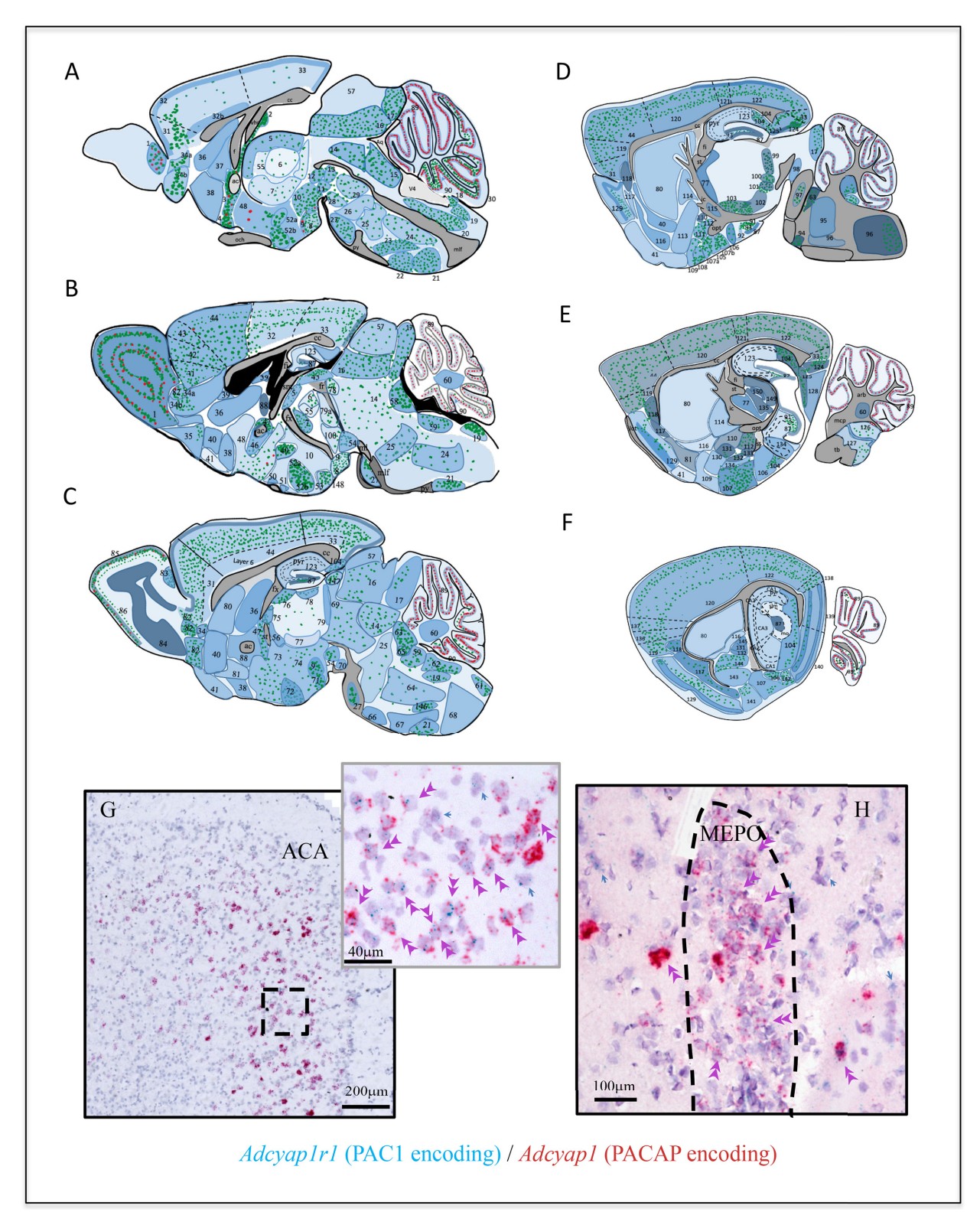

*Adcyap1r1* (PAC1 encoding) / *Adcyap1* (PACAP encoding)

**Figure 3.** *Adcyap1r1* expression assessment in relation to *Adcyap1* expression suggests that the PACAP-PAC1 system uses autocrine, paracrine, and neuroendocrine modes for signal transduction. (**A–F**) Mapping of *Adcyap1r1* (RNA encoding Pac1) expression (symbolized by the intensity of blue shading) in six septo-temporal planes in relation to main PACAP containing brain nuclei and subfields, based on microscopic observations. Green and red dots represent *Adcyap1* expressing neurons of glutamatergic (VGLUTs mRNA expressing) or GABAergic (VGAT mRNA expression) nature,

*Figure 3 continued on next page*

*Figure 3 continued*

respectively. Shaded regions with different blue intensity symbolize the strength of *Adcyap1r1*. For abbreviations see the corresponding table for abbreviations in Appendix 1. 1. MOB; 2. SFO; 3. MEPO; 4. OV; 5. PVT; 6. MD; 7. RE; 8. MBO; 9. SUM; 10. PH; 11. IF; 12. EW; 13. RL; 14. PAG; 15. DR.; 16 SCm; 17. IC; 18. AP; 19. NTS; 20. XII; 21. IO; 22. RPA; 23. RM; 24. GRN; 25. PRNr; 26. tegmental reticular n.; 27. PG; 28. IPN; 29. CLI 30. UVU; 31. ORB; 32. ACAd1; 32b. ACAd6; 33. RSP; 34 DP; 35. TTv; 36. LSr; 37. MS; 38. NDB; 39. LSc; 40. ACB; 41. OT; 42. ILA; 43. PL; 44. MOs; 45. MH; 46. BST.; 47. BAC; 48. MPO; 49. PVH; 50. SCH; 51. SO; 52a. DMH; 52b. VMH; 53. PVp; 54. VTA; 55. AM; 56. PT; 57. SCs; 58. PCG; 59. MV; 60. FN.; 61. CU; 62. SPIV.; 63. PB; 64. IRN; 65. LC; 66. SOC; 67. MARN; 68. MDRN; 69. MRN; 70. SN; 71. LM; 72. VLH; 73. LPO; 74. PHA; 75. AV; 76. AD; 77. RT; 78. PRC; 79. PF; 80. CP; 81. SI; 82. AON.; 83. AOB; 84. MOBgr; 85. MOBgl; 86. MOBml; 87.DG-gcl; 88. BSTov; 89. CBpj; 90. CBgcl; 91. CA3vv; 92. CA3v, 93. vhil; 94. MoV; 95. sV; 96. spV; 97. NLL; 98. PBG; 99. MG; 100. SPF; 101. PP; 102. ZI; 103. STN; 104. SUB; 105. MEApv; 106. PA; 107a. COAa; 107b. COAp; 108. NLOT; 109. AAA; 110. CEAm; 111. MEAad; 112. MEpd; 113. SI; 114. GPe; 115. GPi (entopeduncular nucleus); 116. FS; 117. EP; 118. CLA; 119. AI; 120. SS; 121. PTLp; 122. VIS; 123. DH; 124. POST; 125. PRE; 126. DCO; 127. VCO; 128. PAR; 129. PIR; 130. TT; 131. CEAc; 132. IA, 133. BMA; 134. CA2v; 135. LGv; 136. GU; 137. VISC; 138. ECT; 139. ENTl; 140. ENTm; 141. PAA; 142. TR; 143. BLA; 144. LA 145. CEAl 146. AMB; 147. OP; 148. PM.; 149. LGd; 150. IGL; 151. FL; 152. AN. Aq: aqueduct; och: optic chiasm; v4: forth ventricle; mlf: medial longitutinal fasciculus; cc: corpus callosum; vhc: ventral hippocampus commissure; fi/fx: fimbria/fornix; pyr: pyramidal layer; lot: lateral olfactory tract, mcp: middle cerebellar penducle; st: stria terminalis; opt: optic tract; ic: internal capsule; tb: trapezoid body; arb: arbor vidae. (G and H) Examples illustrating *autocrine* and *paracrine* features of PACAP-PAC1 signaling that *Adcyap1r1* was expressed in PACAP containing (*Adcyap1* expressing) neurons. (G) ACA in prefrontal cortex and (H) MEPO. Double arrowheads indicate co-expression and blue arrows indicate the *Adcyap1r1* expressing neurons which are not *Adcyap1* expressing but were adjacent to them.

The online version of this article includes the following source data and figure supplement(s) for figure 3:

**Source data 1.** DISH mapping of Vipr1 co-expression with VGAT in selective brain regions.
**Source data 2.** DISH mapping of Vipr2 co-expression with VGAT in selective brain regions.
**Source data 3.** Density of VipR1 mRNA expressing cells in the mouse brain: analysis of VGAT mRNA co-expression and comparison with data from Allen Brain Atlas.
**Source data 4.** Density of VipR2 mRNA expressing cells in the mouse brain: analysis of VGAT mRNA co-expression and comparison with data from Allen Brain Atlas.
**Source data 5.** Density distribution of PAC1 expressing cells in selective cortical regions.
**Figure supplement 1.** Examples illustrating PAC1 mRNA (*Adcyap1r1*) was highly co-expressed in cortical and subcortical GABAergic neurons.
**Figure supplement 2.** Neurochemical and anatomical divergence between the distribution of PACAP containing circuits in phylogenetically old and new brain areas were compared, revealing possible evolutionary divergence of PACAP function in mouse and rat.

distant granule cells, depending on whether they project directly to granule cells or to local inhibitory interneurons (*Scharfman and Myers, 2012*). However, the net effect of mossy cell loss on granule cell activity is not clear. Interestingly, dentate gyrus has a unique feature: there are two principal populations of glutamatergic cell type, the granule cells and the mossy cells. The former *intensely* expressed *Adcyap1r1* and the latter *intensely* expressed *Adcyap1*, indicating that PACAP/PAC1 signaling may play a pivotal role for granule cell excitability.

A second population of *Adcyap1*-expressing neurons in ventral hippocampus was *a subset of CA3 pyramidal neurons in the ventral tip*, and have been addressed in the literature as CA3vv pyramidal neurons expressing the gene *coch* (*Thompson et al., 2008*; *Fanselow and Dong, 2010*). This population was previously photo-documented without comment in Figure 11J of the referenced report (*Hannibal, 2002*). This represents a distinct and novel group of pyramidal neurons in ventral CA3 (*Figure 4A, C and D*). These neurons strongly expressed *Adcyap1*, and co-expressed Slc17a7 (*Figure 4*, inset of B and D) as well as *Calb2* (*Figure 4B*), with the rest of pyramidal neurons expressing *Slc17a7* but neither *Calb2* nor *Adcyap1*.

Retrohippocampal regions expressing *Adcyap1* and either *Slc17a7* or *Slc17a6* were entorhinal area, prominently in the layer 5, parasubiculum, postsubiculum, presubiculum, and subiculum (*Figure 3E and F*, areas 139 and 140, 128, 124, 104). This latter region, subiculum, together with the pyramidal layer of dorsal CA1, CA2 and CA3, exhibited large differences in *Adcyap1* expression strength between rat and mouse (*Figure 3—figure supplement 2*). Developmental studies of these regions, as well as extended amygdala, may indicate a recapitulation of phylogeny by development that is relevant to the evolution of PACAP neurotransmission across mammalian species (*Zhang and Eiden, 2019*).

## Cerebral cortex: structures derived from cortical subplate

The subplate is a largely transient cortical structure that contains some of the earliest generated neurons of the cerebral cortex and has important developmental functions to establish intra- and extracortical connections (*Bruguier et al., 2020*). The concept of the subplate zone as a transient,

**Table 2.** Distribution, cell types, and strength of main PAC1 expressing group in mouse cerebral nuclei (striatum and pallidum).

| Cell group / sub-field | Slc17a7 (VGLUT1) | Slc17a6 (VGLUT2) | Slc17a8 (VGLUT3) | Slc32a1 (VGAT) |
|---|---|---|---|---|
| Striatum | | | | |
| Caudoputamen | | | | |
| Nucleus accumbens | - | - | - | +++ |
| Fundus of striadum | - | - | - | ++ |
| Olfactory tubercle | - | - | - | + |
| Lateral septum complex | - | - | - | ++++ |
| Medial amygdala (MeA) | | | | |
| MeAav | - | ++++ | - | + |
| MeApd | - | + | - | ++++ |
| Central amygdala (CeA) | | | | |
| CeA medial | - | - | - | +++ |
| CeA lateral | - | - | - | +++ |
| CeA capsular | - | - | - | ++++ |
| Anterior amygdala area | - | - | - | +++ |
| Intercalated nucleus | - | - | - | ++++ |
| Cell group / sub-field | Slc17a7 (VGLUT1) | Slc17a6 (VGLUT2) | Slc17a8 (VGLUT3) | Slc32a1 (VGAT) |
| Pallidum | | | | |
| Globus pallidum internal | - | - | - | ++ |
| Globus pallidum external | - | - | - | ++ |
| Globus pallidum ventral (VP) | | | | |
| Substantia innominata | - | - | - | ++ |
| Magnocellular nucleus | - | - | - | +++ |
| Medial septal complex | - | - | - | +++ |
| Bed nuclei stria terminalis (BNST) | | | | |
| BNSToval | - | - | - | ++++ |
| BNSTam | - | - | - | +++ |
| BNSTdm | - | - | - | +++ |
| BNSTpr | - | - | - | +++ |
| Nucleus of Diagonal Band | - | - | - | ++++ |
| Bed nucleus of anterior commissure | + | + | - | - |

Semiquantitative annotations are used here the percentage of expressing cell/total Nissl stained nuclei: '-', not detectable; '+'' weak (<20%); '++', low (40–20%); '+++', moderate (60–40%); '++++', intense (80–60%); '+++++', very intense (>80).

dynamically changing and functional compartment arose from the combined application of functional and structural criteria and approaches (for a historical review see *Judaš et al., 2010*). Here, we adapt our classification from that of the Allen Brain Map (https://portal.brain-map.org/). Two noteworthy structures derived from the cortical subplate that expressed *Adcyap1* and *Adcyap1r1* are the claustrum (CLA) and the lateral amygdalar nucleus (LA).

## The claustrum (CLA)

Owing to its elongated shape and proximity to white matter structures, the claustrum (CLA, *Figure 3*, panels D, E, F, area 118) is an anatomically well-defined yet functionally poorly described structure, once speculated to be the 'seat of consciousness' due to its extensive interconnections (*Crick and Koch, 2005*). CLA is located between the insular cortex and the striatum: it is a thin sheet of gray matter considered as a major hub of widespread neocortical connections (*Bruguier et al., 2020*).

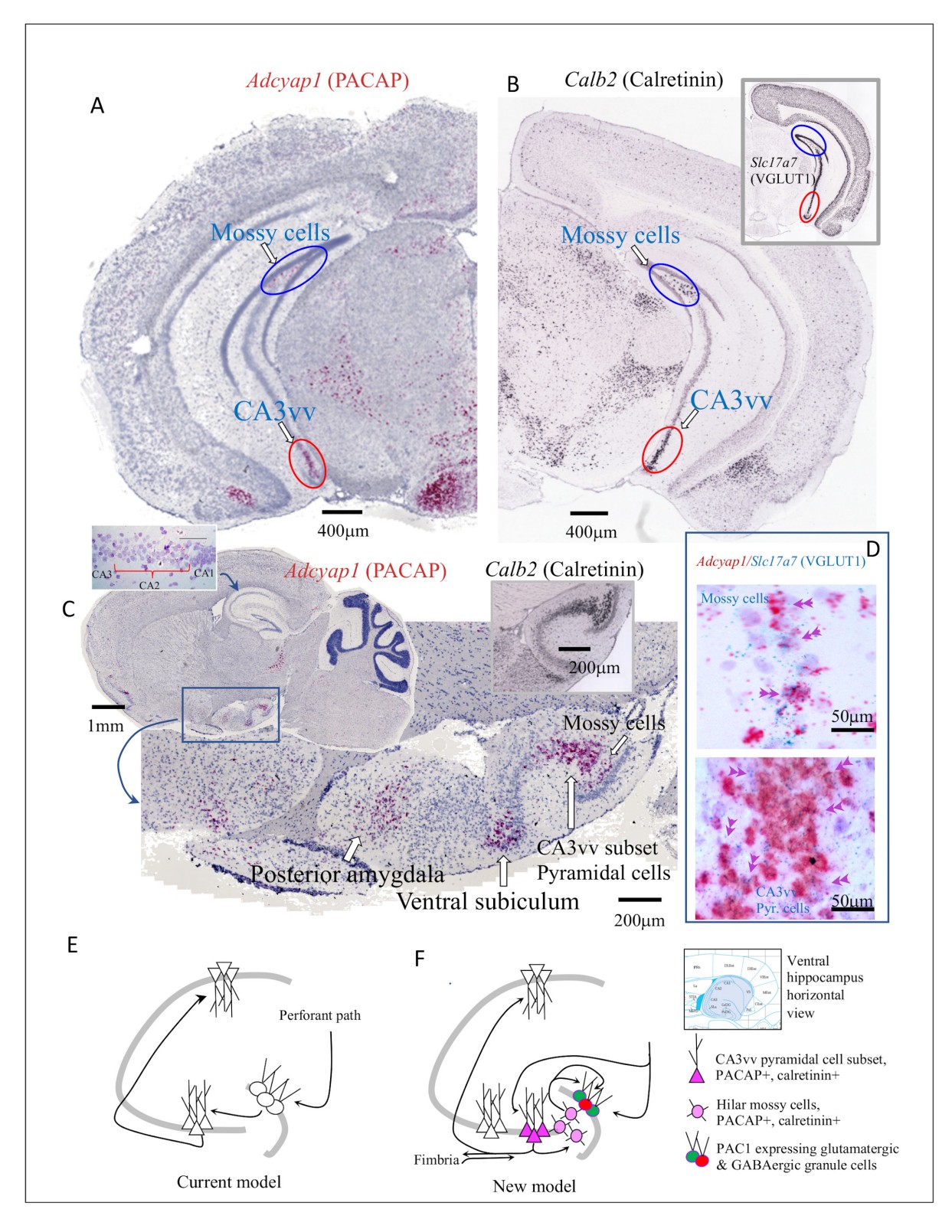

**Figure 4.** Ventral (temporal pole) hippocampus CA3 (CA3vv) contained a newly identified subset of pyramidal neurons distinguished by its molecular signatures of VGLUT1, PACAP and calretinin mRNA expression. Low-magnification bright field coronal (**A**) and sagittal (**C**) whole-brain sections with ISH (RNAscope 2.5 High Definition (HD) Red Assay), showing the selective expression of mRNA of PACAP (*Adcyap1*) in a subset of CA3 of the temporo-ventral pole of hippocampus (red ovals circumscribed regions in A and C). Hilar mossy cells in the dorsal (**A**) and ventral (**C**) hippocampus, also *Adcyap1*

*Figure 4 continued on next page*

*Figure 4 continued*

expressing, are circumscribed by blue ovals. (**B**) and inset show the corresponding coronal sections in low magnification of *Calb2* (calretinin) and *slc17a7* (VGLUT1) taken from Allen Brain Atlas (*Ng et al., 2009*) where CA3vv subset and hilar mossy cells are indicated with red and blue ovals, respectively). Right inset of C corresponds to calretinin mRNA expression in the same hippocampus sagittal squared region of C. Both the subset of CA3vv pyramidal neurons and the mossy cells co-expressed VGLUT1 mRNA (*Slc17a7*) and *Adcyap1* (**D**). Left inset of C shows dorsal CA2 pyramidal layer expressed *Adcyap1*. The 'trisynaptic-centric' (**E**) vs 'CA3-centric' (**F**) view of hippocampal information processing, where the newly identified CA3vv subset of *Adcyap1* and *Calb2* containing glutamatergic neurons are presented in dark pink triangles and the mossy cells are in light pink circles. The *Adcyap1r1* expressing granule cells (green) and interneurons (red) in the granule cell layer are symbolized with pink circle. Chartings were based on ventral pole of hippocampus (shaded region of atlas segment (*Paxinos mouse brain*), where this chemically distinct subset of CA3c pyramidal neurons was identified. Circuits were modified from *Scharfman, 2007*, with adaptation to the new finding from this study.

The CLA is recently reported to be required for optimal behavioral performance under high cognitive demand in the mouse (*White et al., 2020*). Consistent with recent work (*White et al., 2018*), rat CLA receives a dense innervation from the anterior cingulate cortex (ACA), one of the most prominent PACAP mRNA-expressing regions in frontal cortex, co-expressing *Slc17a7* and *Slc17a6* (*Table 1* and *Figure 3A*, areas 32a and 32b, and G) and is implicated in top-down attention (*Zhang et al., 2016b*). The CLA interconnects the motor cortical areas in both hemispheres through corpus callosum (*Smith and Alloway, 2010*), where EGFP+ projections were reported in PACAP promoter-EGFP reporter mice (*Condro et al., 2016*). Expression of the neuropeptides somatostatin (SOM), cholecystokinin (CCK), and vasoactive intestinal polypeptide (VIP) has been reported in the rat CLA (*Eiden et al., 1990*). In mouse CLA, more than 80% of the neurons were *Slc17a7*- and *Slc17a6*-coexpressing and less that 20% of the neurons expressed *Slc32a1*. PACAP content in rodent CLA has not been reported. In our study, we observed 10–15% of glutamatergic cells of the CLA co-expressed *Adcyap1* and almost 100% of cells expressed *Adcyap1r1* (*Figure 3D*, area 118).

## Endopiriform nucleus and amygdalar complex

The endopiriform nucleus (*Figure 3E* and F, area 117) and divisions of lateral (*Figure 3F*, area 144), basolateral (*Figure 3F*, area 143), basal medial (*Figure 3E*, area 133), and posterior amygdalar (*Figure 3F*, area 106, *Figure 4C*) nuclei are, from a phylogenetic point of view, olfactory structures (*Groor, 1976*) derived from cortical subplate which the main cell population is glutamatergic, co-expressing *Slc17a7* and *Slc17a6* (see *Table 1*). *Adcyap1* was *intensely* expressed in the lateral (dorsal) amygdala (*Figure 3F*, area 144), anterior basomedial amygdala (*Figure 3E*, area 133), posterior amygdalar nucleus (area 106), and with *low* expression in the endopiriform nucleus (area 117) and basomedial amygdala posterior subnucleus (*Figure 1—source data 1G*) and *weak* expression in the basolateral amygdala (*Figure 3F*, area 143).

## Structures derived from cerebral nuclei

The main structures expressing *Adcyap1* in striatum were the lateral septum (LS, *Figure 3B*, areas 36 and 39) and medial amygdala (MEA, *Figure 3D*, areas 111 and 112) (SI *Figure 1G*). Most of these neurons coexpressed both *Slc17a7* and *Slc17a6* (see *Table 1*). In contrast to rat brain, where expression of *Adcyap1* is prominent in central amygdala and intercalated cells (*Figure 3—figure supplement 2, C*), in the mouse *Adcyap1*-expressing cells were quite sparse in these structures (*Figure 3—figure supplements 2C'*). *Adcyap1r1* was *intensely* expressed in these mainly *Slc32a1*-expressing structures (*Figure 2D,E,F and G*, *Table 2*).

In structures within pallidum, *Adcyap1* was expressed, in order of abundance, in posterior, and anterior divisions of BST, and very *weakly*, in the oval nucleus (*Table 1*).

The *very intense* expression of *Adcyap1r1* was observed on mainly GABAergic structures derived from cerebral nuclei. *Figure 2D and G* show examples illustrating *Adcyap1r1*- and Slc32a1-coexpressing neurons in some subcortical regions. Most of cells in the BST complex co-expressed *Adcyap1r1* (*Figure 2D and E*). In BSTov, we tested the three main GABAergic cell types that co-express somatostatin (Sst), parvalbumin (Pvalb), and corticotropin releasing hormone (Crh) and found all the three types of neurons co-expressed *Adcyap1r1* (*Figure 3—figure supplement 1, I,J and K*). *Table 2* summarizes the distribution, cell types, and strength of expression of the main *Adcyap1r1*-expressing group in mouse cerebral nuclei.

The bed nucleus of anterior commissure (BAC) is defined here, for the first time, as a major *Adcyap1*-expressing nucleus (**Figure 1C**, **Figure 3B**, area 47, **Figure 5A**, F, Si **Figure 1E**). BACs are bilateral triangular cell groups located on the dorsal corners of the *chiasm* of anterior commissure (ac) between the anterior part and posterior part (**Papp et al., 2014**). Most BAC neurons (90–95%) express *Slc17a6 intensely* (**Figure 5B**) and *Slc17a7* weakly (**Figure 5C**) and around 5% express *Slc32a1* (**Figure 5D** and inset). *Calb2* is *intensely* expressed in BAC (**Figure 5E**), as in the case of the subset of CA3vv pyramidal neurons mentioned earlier (**Figure 4A, C**). The *Adcyap1*-expressing neurons were densely packed (**Figure 1**, C and **Figure 5A**). Almost all the *Adcyap1*-positive neurons co-expressed *Slc17a6* mRNA (**Figure 5F**) and *Slc17a7*. Co-expression of *Adcyap1* within the Slc32a1-positive neurons was not found in BAC (**Figure 5G**).

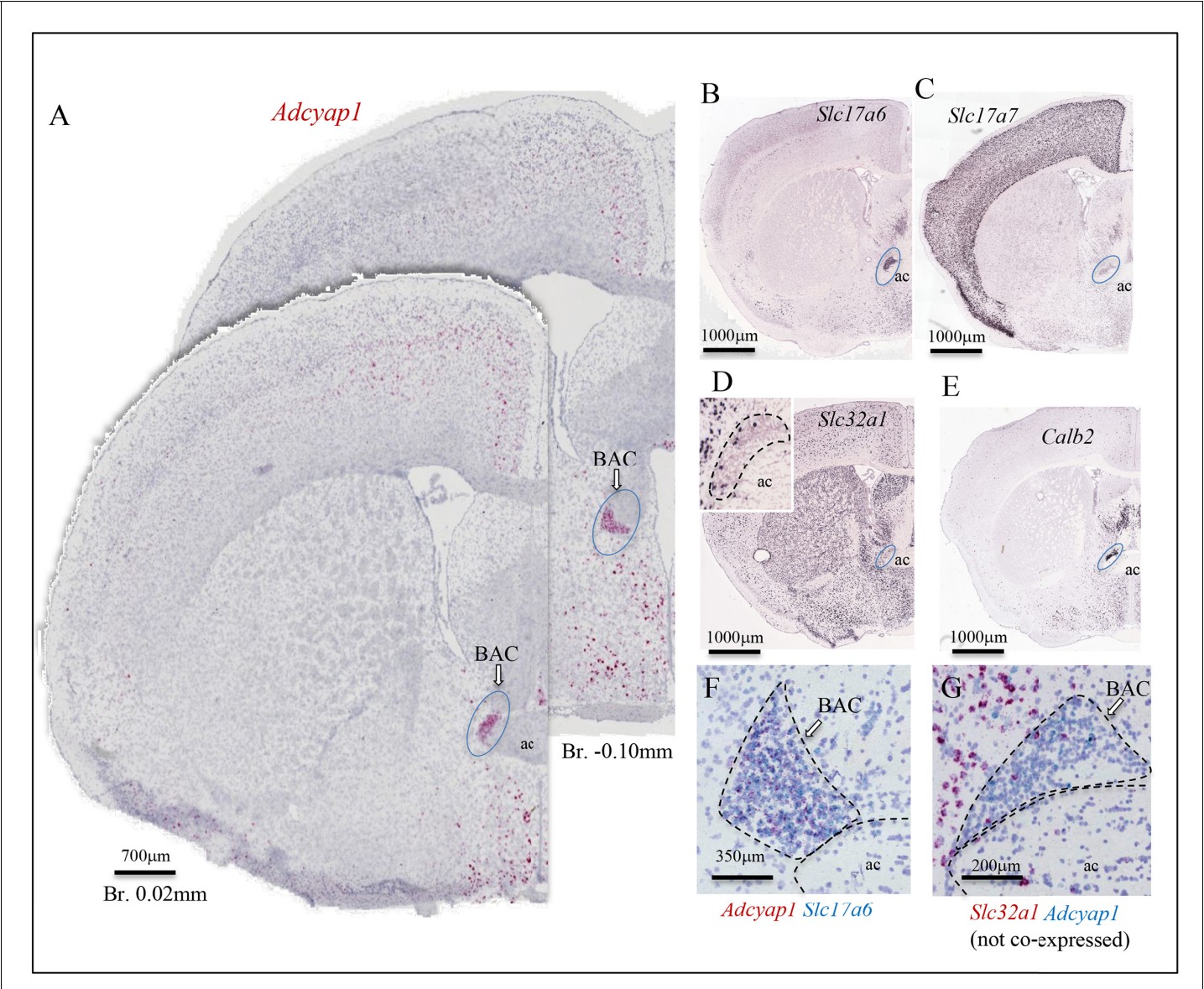

**Figure 5.** Bed nucleus of anterior commissure (BAC): a prominent PACAP containing glutamatergic nucleus chemo-anatomically identified. (**A**) Two coronal sections at Bregma 0.02 mm and −0.10 mm of mouse brain showing *Adcyap1* ISH (RNAscope 2.5 High Definition (HD) Red Assay) expressing BAC (ac: anterior commissure). Panels B–E are low-magnification photomicrographs taken from Allen Brain Atlas (**Ng et al., 2009**) showing the *Slc17a6* (VGLUT2, (**B**)) *Slc17a7* (VGLUT1, (**C**)) *Slc32a1* (VGAT, (**D**) and inset), *Calb2* (calretinin, (**E**)) expressed in BAC. The *Adcyap1* expressing neurons were densely packed and co-expressed *Slc17a6* (**F**) and we did not observe co-expression within the *Slc32a1*-expressing cells (**G**).

## Brain stem

### Interbrain

### Thalamus

*Adcyap* is extensively expressed in thalamic nuclei (see *Table 1*), most prominently in suprafasicular nucleus (SPF, SI Figure 1I and I3), paraventricular nucleus of the thalamus (PVT, *Figure 3A*, area 5), medial geniculate complex (*Figure 3D*, area 99), the nucleus reuniens (*Figure 3A*, area 7), and the mediodorsal thalamic nucleus (MD) (*Figure 3A*, area 76) neurons co-expressing *Slc17a6*. The medial geniculate nucleus or medial geniculate body is part of the auditory thalamus and represents the thalamic relay between the inferior colliculus and the auditory cortex. The nucleus reuniens receives afferent input mainly from limbic and limbic-associated structures, mediating interactions between the hippocampus and medial prefrontal cortex important for spatial working memory (*Griffin, 2015*). It sends projections to the medial prefrontal cortex, the hippocampus, and the entorhinal cortex (*Wouterlood et al., 1990*; *McKenna and Vertes, 2004*), although there are sparse connections to many of the afferent structures as well. The prefrontal cortical-hippocampal connection allows regulation of neural traffic between these two regions with changes in attentiveness (*Vertes et al., 2007*) as well as in resilience to stress (*Kafetzopoulos et al., 2018*). All the thalamic nuclei that express *Adcyap1* also express *Adcyap1r1* (*Table 1* and *Figure 3A–E*). The PVT and MD participate in many sensory information relays. In a recent study, their role in a key neural circuit for psychological threat-induced hyperthermia was reported (*Kataoka et al., 2020*). This circuit involves brain regions in the prefrontal pole, called the dorsal peduncular area (DP, *Figure 3A and B*, 34a, also called dorsal taenia tecta, TTd, a main *Adcyap1*-containing region mentioned in section The claustrum) that senses social stress and mediates increased body temperature in response to it (*Lin, 2020*). Neurons from the DP/TTd then project to and excite neurons in the dorsomedial hypothalamus (DMH, another *Adcyap1*-containing nucleus in hypothalamus, *Table 1*, *Figure 3B*, area 52a, and *vide infra*), which in turn sends neuronal projections to the rostral medullary raphé (rMR, also a *Adcyap1*-expressing nucleus, *Table 1*, *Figure 3A*, area 23, and description vide infra).

### Epithalamus: habenula

Habenulae are bilateral triangular eminences of the stalk of the pineal gland, situated at the dorso-caudal end of the thalamus. Their medial divisions border the third ventricle. The habenula is considered as the relay hub where incoming signals from basal forebrain, including, diagonal band of Broca, lateral preoptic area, lateral hypothalamus, paraventricular nucleus, and entopeduncular nucleus, travel through the stria medullaris to habenula to be processed. The habenula then conveys the processed information to midbrain and hindbrain monoaminergic structures, such as ventral tegmental area, medial and dorsal raphe nuclei, and periaqueductal grey, through the fasciculus retroflexus. The habenula thus connects the cognitive-emotional basal forebrain to the modulatory monoaminergic area (*Sutherland, 1982*). Medial habenula (MHb, *Figure 3B*, area 45), was observed to express strongly *Adcyap1* in the dorsal half, in cells which co-express *Slc17a7* or *Slc17a6* (*Figure 1—source data 1G*). In the lateral habenula (*Figure 3B*, area 78), the *Adcyap1*-positive neurons co-expressed *Slc17a6*, and were mainly located in the central nuclei of the lateral habenula (*Figure 1—source data 1G*), a region with rich input from hypothalamic peptidergic afferents including arginine vasopressin and orexin (*Zhang et al., 2018*). All those cells, both in lateral and medial habenula, co-expressed *Calb2* (see Allen Brain Atlas for reference https://mouse.brainmap.org/experiment/show?id=79556662).

### Hypothalamus

Using the sensitive DISH method, a total of 26 hypothalamic nuclei were found to express *Adcyap1* (*Table 1*). Among the highest density *Adcyap1*-expressing cell clusters/nuclei (>80% of cells *Adcyap1*-positive) of hypothalamus are (numbers refer to *Figure 3A, B, C*): SFO (2, *Figure 1—source data 1F*), MEPO (3, *Figure 1—source data 1C, D*), OVLT (4, *Figure 1—source data 1C*), DMH (52a) and VMH (52b, *Figure 1—source data 1G*), STN (*Figure 3D*, 103, *Figure 1—source data 1H*), and lateral mammillary nucleus (71, *Figure 1—source data 1I*). All these *Adcyap1*-positive expressing cells co-expressed *Slc17a6*. Other hypothalamic regions listed in *Table 1* had lower density of expression and were also *Slc17a6*-positive co-expressing, such as paraventricular hypothalamic nucleus (PVH, area 49, *Figure 1—source data 1F*). *Slc32a1/Adcyap1*-coexpressing cells were

sparsely distributed mainly in the anterior hypothalamic area (AHA *Figure 1—source data 1O3*), supramammillary (*Figure 1—source data 1O4*), and tuberomammillary nuclei (*Figure 1—source data 1O2*).

*Adcyap1r1* expression in hypothalamus was extensive (*Figure 3A–C*) and in fact ubiquitous. The nuclei with intense *Adcyap1* expression mentioned above also had intense expression of *Adcyap1r1*. In addition, the PVH (49), SO (51), SCH (50), DMH (52a), arcuate hypothalamic, anterior hypothalamic nucleus, zona incerta (102, *Figure 1—source data 1H*), postero-lateral hypothalamic area (LH, area 74), periventricular hypothalamic nucleus posterior, dorsal premammillary nucleus and supramammillary nucleus medial (*Figure 1—source data 1I and I10*) exhibited strong *Adcyap1r1* expression. Both Slc32a1- and Slc17a6-positive cells co-expressed *Adcyap1r1*.

## Midbrain

We found moderate expression of *Adcyap1* with *Slc17a6* in the mainly sensory-related structures: inferior colliculus (IC, *Figure 3A, B, C*, area 17), nucleus of the brachium of IC, midbrain trigeminal nucleus, and sparse expression (between 20–40%) in parabigeminal nucleus (*Table 1*, *Figure 3D*, area 98). We found *intense* co-expression of *Adcyap1* and *Slc17a6* in motor related structures: superior colliculus, motor related subfield (SCm, *Figure 3A–C*, area 16, *Figure 1—source data 1I, J, K*) and periaqueductal gray (PAG, *Figure 3A*, area 14, *Figure 1—source data 1I-K*); *moderate* expression in ventral tegmental area (VTA, *Figure 3B*, area 54, *Figure 1—source data 1I and I9*) and Edinger-Westphal nucleus (*Figure 3A*, area 12). *Adcyap1r1* was strongly expressed in the PAG and SC, although a moderate expression of this transcript was observed to be widespread in *Slc17a6*- and *Slc32a1*-expressing cells (*Figure 3A–D*).

## Hindbrain

### Pons

In sensory-related structures, we found *intense Adcyap1* expression in the parabrachial complex (PBC, *Figure 3C*, area 63), in all its subfields, although it was more intense toward its lateral divisions, external to the superior cerebellar peduncles (scp, *Figure 1D* and *Figure 1—source data 1K, L*). The cells in those divisions, except the Koelliker-Fuse subnucleus (KF), were small cells mainly co-expressing *Slc17a6*. In contrast, the *Adcyap1*-positive cells in KF were bigger than the cells in the rest of the PBC divisions and strongly co-expressed *Slc17a7* (*Figure 1D*). Other structures with moderate expression of *Adcyap1* and *Slc17a7* were the lateral division of SOC (*Figure 3C*, area 66 and *Figure 1—source data 1L*) and DCO and VCO (*Figure 3E*, area 126 and 127, *Figure 1—source data 1K, M, M6*). Lateral leminiscus nucleus and the principal sensorial nucleus of the trigeminal (*Figure 3D*, area 95, *Figure 1—source data 1L*) expressed *Adcyap1 moderately* in *Slc17a6*-expressing cells. In motor-related structures, we found *moderate Adcyap1* co-expressed with *Slc17a6* in the following structures: tegmental reticular nucleus, Barrington's nucleus, dorsal tegmental nucleus, pontine grey (PG, *Figure 3A*, area 27, *Figure 1—source data 1J*), pontine reticular nucleus (*Figure 3A*, area 25, *Figure 1—source data 1K, L*), supratrigeminal nucleus (SUT, *Figure 1—source data 1L*), and superior central nucleus raphe (*Table 1* and *Figure 3A–C*). In behavioral-state-related structures, we found *intense Adcyap1* expression in locus coerulus (LC) neurons co-expressing *Slc17a7* or *Slc17a6*. The laterodorsal tegmental nucleus also expressed *intensely Adcyap1*, co-expressed with *Slc17a6*. Pontine reticular nucleus and superior central nucleus of raphe expressed *Adcyap1* with *Slc17a6* in a *moderate* manner (*Table 1*).

*Adcyap1r1* was strongly expressed in pons structures, which also *intensely* expressed *Adcyap1*, such as PG, LC, PBC, DTN regions. Otherwise, the expression was observed to be widespread in both glutamatergic and GABAergic cell types (*Figure 3A–D*).

### Medulla

In the medulla oblongata, *Adcyap1* was extensively expressed and generally in a sparse pattern. However, some of the nuclei showed strong-intense expression: the nucleus of tractus solitarius (NTS, *Figure 3A*, area 19), medial division (co-expressing mainly *Slc17a6*), the NTS lateral division (co-expressing mainly *Slc17a6* and occasionally *Slc17a7*), and the dorsal and ventral cochlear nuclei (co-expressing Slc17a7 or Slc17a6). Details of other *Adcyap1*-positive neurons co-expressing Slc17a6 nuclei can be found in *Table 1*.

The *Adcyap1r1* expression in medulla is similar to pons, a widespread pattern with *intense* expression in NTS divisions, and other nuclei, which expressed *Adcyap1* (*Figure 3A–D*).

## Cerebellum

In the cerebellum, *Adcyap1* was expressed in all Purkinje cells, which co-expressed *Slc32a1* (*Figure 1*, *Figure 3A–F* and *Figure 1—source data 1M, M1-M4, O and O1*). *Figure 1*, panels E-H show two cerebellar regions, paraflocculus (E and F) and central (G and H), in low and high magnification, respectively, where *Adcyap1* expression was higher than other cerebellar lobules. Purkinje cells, distributed in all regions of cerebellar cortex, are the most prominent population of GABA/PACAP co-expressing neurons of the brain. Some GABAergic cells in the granule cell layer of paraflocculus and central regions also co-expressed *Adcyap1* (indicated with double pink arrowheads in *Figure 1E–H*), although we could not identify whether these were, Golgi, Lugaro or globular cells (*Simat et al., 2007*).

In these two cerebellar regions, some granule cells also expressed *Adcyap1* (indicated with single blue arrows, *Figure 1E–H*). In deep cerebellar nuclei, few *Adcyap1*-expressing cells were found in fastigial, interposed and dentate nuclei, with the former two co-expressing Slc17a7 and the latter co-expressing Slc17a6 (*Table 1*). *Adcyap1r1* expression here was more limited than in other brain regions analyzed above. *Adcyap1* was mainly expressed in the Slc32a1-expressing Purkinje cells and sparsely expressed in Slc17a7- and Slc17a6-expressing neurons in the deep cerebellar nuclei (*Figure 3A–F*).

## PACAP→PAC1 signaling within sensory and behavioral circuits

Here, we analyze the chemo-anatomical aspects of PACAP/PAC1 mRNA expression using the results described above, but putting them into basic sensory circuit wiring maps, as well as in behavioral state and survival instinctive brain longitudinal structures, especially the hypothalamic hubs, based on existing classification schema (*Swanson, 2012*; *Sternson, 2013*; *Swanson et al., 2016*; *Zimmerman et al., 2017*; *Swanson, 2018*). Based on PACAP mRNA expression in these proposed sensory/behavioral circuits, we have addressed consequences of PACAP deficiency on neuronal activation and behavioral output in a mouse model of predator odor exposure and defensive behavior.

## PACAP-PAC1 co-expression in forebrain sensory system
### Thirst circuit for osmotic regulation

As shown above, *Adcyap1* was intensely expressed in all peri/para ventricular structures directly related to thirst and osmotic regulation (Figure 7). These structures include SFO, OVLT, MEPO, and PVH (vide supra). Other hypothalamic nuclei intrinsically related to osmotic control and anticipatory drinking are SO and SCH, which were *intensely Adcyap1r1*-expressing.

The SFO is an embryonic differentiation of the forebrain roof plate, in a dorsal region between the diencephalon (interbrain, thalamus) and the telencephalon (endbrain) (*Swanson and Cowan, 1979*; *Anderson et al., 2001*). This nucleus lacks a normal blood-brain barrier, and so its neurons are exposed directly to peptide hormones in the blood. One such hormone is angiotensin II, whose blood levels are elevated upon loss of body fluid due to dehydration or hemorrhage. Hence, the SFO is a *humorosensory organ* that detects hormone levels in the circulation to control drinking behavior and body water homeostasis. The SFO is situated immediately dorsal to the third ventricle and contains intermingled populations of glutamatergic (VGLUT2-PACAP-PAC1) and GABAergic (VGAT-PAC1) neurons with opposing effects on drinking behavior. Optogenetic activation of SFO-GLUT neurons stimulates intensive drinking in hydrated mice, whereas optogenetic silencing of SFO-GLUT neurons suppresses drinking in dehydrated mice (*Zimmerman et al., 2017*). By contrast, optogenetic activation of SFO-GABA neurons suppresses drinking in dehydrated mice (*Bichet, 2018*). SFO-GLUT projections to the median preoptic nucleus (MEPO) and OVLT drive thirst, whereas SFO-GLUT projections to the ventrolateral part of the bed nucleus of the stria terminalis (BSTvl) promote sodium consumption (*Zimmerman et al., 2017*). SFO-GLUT projections to the paraventricular (PVH) and supraoptic (SO) nuclei of the hypothalamus *Anderson et al., 2001* have not yet been functionally annotated with cell-type specificity, but classic models suggest that these projections mediate secretion of arginine vasopressin (AVP) and, in rodents, oxytocin (OXT) into the circulation by posterior pituitary (PP)-projecting magnocellular neurosecretory cells (MNNs). Recent studies also

demonstrated that these MNNs possess ascending projections innervating limbic structures such as amygdala, hippocampus, lateral habenula, and lateral hypothalamus (*Hernández et al., 2015*; *Hernández et al., 2016*; *Zhang et al., 2021*) in a cell-type specific manner (*Zhang and Hernández, 2013*; *Zhang et al., 2016a*; *Zhang et al., 2018*). When these are activated, their central collaterals can exert motivational effect on exploration and drinking behavior.

Thirst and AVP release are regulated not only by the classical homeostatic, interosensory plasma osmolality negative feedback (through SFO as a *humorosensory organ*), but also by novel, extero-sensory, anticipatory signals (*Gizowski et al., 2016*). These anticipatory signals for thirst and vaso-pressin release converge on the same homeostatic neurons of circumventricular organs that monitor the composition of the blood. Acid-sensing taste receptor cells (which express polycystic kidney disease 2-like one protein) on the tongue that were previously suggested as the sour taste sensors also mediate taste responses to water. Recent findings obtained in humans using blood oxygen level-dependent (BOLD) signals demonstrating that the increase in the lamina terminalis (LT) BOLD signal observed during an infusion of hypertonic saline is rapidly decreased after water intake well before any water absorption in blood. This is relevant in the context of this paper since the MEPO of the hypothalamus has been shown to mediate this interesting phenomenon, integrating multiple thirst-generating stimuli (*Allen et al., 2017*; *Gizowski and Bourque, 2017*); however, functionally anno-tated cell-type specific circuitry has not been clarified. *Very intense* expression of *Adcyap1* was observed in MEPO. Together, these observations open new possibilities to further understand the role of PACAP-PAC1 signaling within this nucleus for homeostatic and allostatic control.

Information about plasma sodium concentration enters the circuit through specialized aldoste-rone-sensitive neurons in the NTS, an *intense* PACAP-expressing nucleus in the medulla, that expresses 11β-hydroxysteroid dehydrogenase type 2 (NTS-HSD2 neurons) (*Zimmerman et al., 2017*; *Bichet, 2018*), which promote salt appetite and project to the LC and PBN, both *intense Adcyap1*-expressing and BSTvl, *moderate Adcyap1*-expressing and *intense Adcyap1r1*-expressing (*Figure 6A*, basic circuit presented based on above literature).

## Olfactory pathway

In the main olfactory system, input from olfactory sensory neurons reaches the main olfactory bulb (MOB), and the axons of the projection neurons in MOB travel to the anterior olfactory nucleus (AON), piriform cortex (Pir), and amygdala for additional processing (*Wacker et al., 2011*). This multi-step pathway brings olfactory information to be processed in multiple areas of the cerebral cortex, including the anterior AON and Pir. The AON, a cortical area adjacent to the olfactory bulb, is part of the main olfactory pathway. A parallel system, the accessory olfactory system, brings infor-mation, for instance that conveyed by pheromones, from the vomeronasal organ into the accessory olfactory bulb, which innervates the MEA, BST, and cortical amygdala (Figure 10). Although it was originally assumed that only the accessory olfactory system processed pheromonal and other socially relevant odors, more recent evidence suggests that social information is processed by both path-ways (*Wacker and Ludwig, 2012*).

The olfactory system appears specially to use PACAP/PAC1 as one of its main modes of co-trans-mission (see section The claustrum), especially within the cell population in the outer plexiform layer of the MOB and in the mitral cell layer of the AOB, in which some cells were observed to co-express *Adcyap1* and *Slc17a7*, *Slc17a6*, or *Slc32a1* (*Figure 1—source data 1A2*) and *Figure 1A–F*.

## Visual pathway and the circadian circuits for brain states

There is a vast literature on the visual system and we only touch on selected hubs here to emphasize the PACAP-PAC1 signaling role for visual information processing (*Figure 6C*). PACAP-PAC1 in the visual system was first discovered via identification of the PACAP immunopositive RGCs, soon after the discovery of PACAP itself (*Hannibal et al., 1997*), which project to the SCH. Here, using DISH method, we report *Adcyap1* co-expressed with *Slc17a6* in retina ganglion cells. *Adcyap1* expression intensity followed a circadian oscillation mode (see section Retina). The RGC sends visual information through its axons forming the optic nerve, chiasm and optical tract and accessory optical tract, with its first offshoot to SCH for brain state modulation. The SCH was found *intensely* expressing *Adcyap1r1* in both *Slc32a1*- and *Slc17a6*-expressing neurons widely and homogenously distributed, at variance with the observation in rat that PAC1 is expressed mainly in the ventral part of the SCH (*Hannibal et al., 1997*). Leaving the optic chiasm, the optic tract courses latero-caudally and emits a

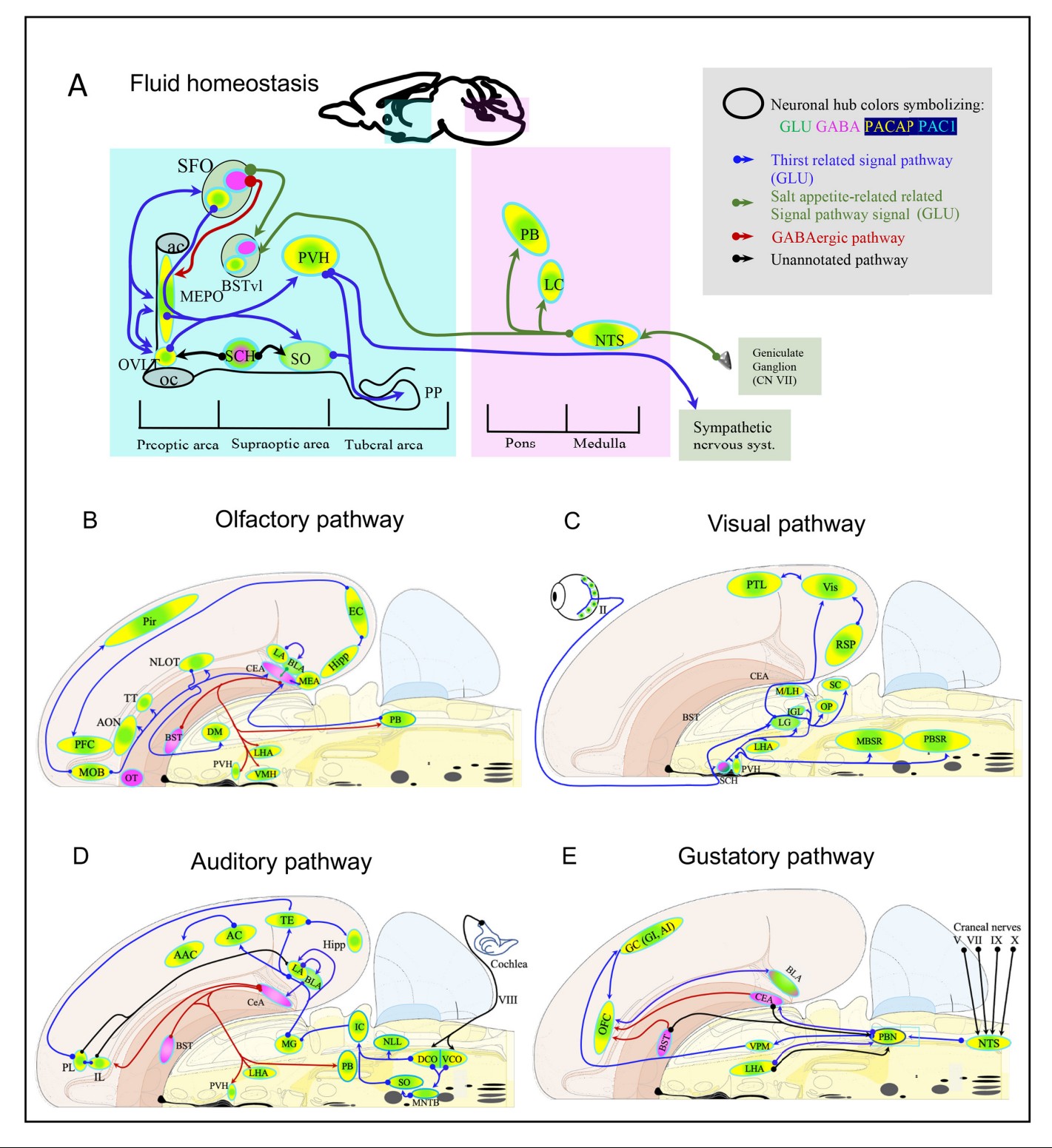

**Figure 6.** Mapping the spatial distribution of PACAP-PAC1 hubs within glutamate/GABA context in relevant sensory circuits in mice. For abbreviations also see the corresponding table in Appendix 1. (**A**) Thirst and salt appetite-related pathways with PACAP-PAC1 glutamatergic / GABAergic signaling noted. The main figure is the enlargement of color-shaded areas of the box in the inset at the upper right, projected against a midsagittal section of mouse brain. Blue shaded area symbolizes the hypothalamus and pink shaded area the hindbrain. (**B**) Olfactory pathway. The projection neurons from the OB send their axons to the different structures of the olfactory cortex, among them AON, TT, OT, PIR, the amygdaline complex (LA, BLA, CEA,

*Figure 6 continued on next page*

*Figure 6 continued*

BST), ENT, and nLOT). (**C**) Visual pathway and circadian circuit for brain states. II: optic nerve; PVN: paraventricular nucleus; LHA: lateral hypothalamic area; LG: lateral geniculate nuclei; IGL: intergeniculate leaflet; M/LHb: medial and lateral habenula; OP: olivary pretectal nucleus, SC: superior colliculus; Vis: visual area; PTL: parietal association area; RSP: retrosplenial area; MBSR: midbrain behavioral state related (pedunculopontine nucleus, substantia nigra, midbrain raphe nuclei; PBSR: pons behavioral state related (locus coeruleus, superior central nucleus of raphe, pontine reticular nucleus). (**D**) Auditory pathway. VIII: cochlear nerve; DCO/VCO: dorsal and ventral cochlear nuclei; MNTB: medial nucleus of the trapezoid body; SO: superior olivary complex; NLL: Nucleus of the lateral lemniscus; IC: Inferior colliculus; MG: medial geniculate complex; AC: Auditory cortex; AAC: associate auditory cortex; TE: temporal association area; Hipp: hippocampus; LHA: lateral hypothalamic area; PVN: hypothalamic paraventricular nucleus; IL: infralimbic cortex; PL: prelimbic cortex. (**E**) Gustative pathway. V, VII, IX, X: represent trigeminal, facial, glossopharyngeal and vagus nerves respectively. NTS: Nucleus of the solitary tract (nucleus of tractus solitarius); NA: nucleus ambiguus; PBN: Parabrachial nucleus; VPM: ventropostero medial nucleus of the thalamus; BLA: Basolateral amygdala; CeA: central Amygdala; BST: bed nucleus of the stria terminalis; LHA: lateral hypothalamic area; RF: reticular formation.

second offshoot, the accessory optical tract, splits off and courses to the midbrain, where it ends in three terminal nuclei, that is medial, lateral and dorsal terminal nuclei. They play an important role in controlling eye movements and are thus parts of the motor system and were all *Adcyap1*-expressing. The main optic tract continues on to end in the lateral geniculate complex (LG) of the thalamus, after giving off collaterals to the superior colliculus (SC) of the midbrain and to the olivary pretectal nucleus (OP). The dorsal part of the LG then projects to primary visual cortex (Vis), whereas the OP is involved in visual reflex, and the superior colliculus has two main roles: projecting to the motor system, and projecting to secondary visual cortical area via thalamus (*Swanson, 2012*). All above optic nerve/tract/accessory tract targeted structures were observed to have intense expression of *Adcyap1r1*. The visual cortex, parietal associated area, and retrosplenial area *intensely* expressed *Adcyap1* (*Table 1* and *Figure 1—source data 1I, J, K* and *Figure 1A–F*).

*Adcyap1r1* was strongly expressed in SCH (*Figure 6C* and *Figure 7*), which projects in turn to several *Adcyap1r1*-expressing regions, which control brain state (sleep-wake cycle), such as midbrain behavioral state related structures (MBSR: pendunculopontine, midbrain raphe nuclei, all *Adcyap1*-expressing) and pons behavioral state related (PBSR: locus coeruleus, superior central nucleus of raphe, pontine reticular nucleus, all *Adcyap1*-expressing) (*Figure 1A–F*).

## PACAP-PAC1 co-expression in the central processing of ganglion cell sensory systems

The sensory ganglion cells were observed to express PACAP shortly after its discovery (*Sundler et al., 1996*). PACAP mRNA and peptide levels are increased in sensory ganglion cells within 1 day following axotomy, suggesting possible roles in neuronal protection, differentiation, nerve fiber outgrowth, and/or restoration of perineuronal tissue upon neuronal damage. The sensory ganglion cells send their axons to primary sensory nuclei in the dorsal medulla, which include: (a) auditory system, which ends in the cochlear nuclei (*Figure 6D*); (b) vestibular system, which ends in the vestibular nuclei (not included in this analysis); (c) gustatory system which ends in the rostral nucleus of *tractus solitarius* (NTS, *Figure 6E*); and (d) vagal/glossopharyngeal visceroceptive system, which ends in the caudal nucleus of the NTS (not included in this analysis). The special sensory nuclei in the medulla are all derived in the embryo from a highly differentiated, dorsal region of the primary hindbrain vesicle, the rombic lip (Figure 5.14 of *Swanson, 2012*) – they were all observed to co-express *Adcyap1* and *Slc17a6* from *very intense* (NTS) to *moderate* levels (DCN and VCN) (see *Table 1*, *Figure 1A–F*, *Figure 1—source data 1M5-M6* and website).

Here, we analyze the PACAP/PAC1 participation in the auditory and gustatory central processing circuits.

### Auditory pathway

Auditory information, for example the spectrum, timing, and location of sound, is analyzed in parallel in the lower brainstem nuclei, that is, dorsal and ventral cochlear nuclei (DCN and VCN), medial nucleus of trapezoid body (MNTB), superior olivary complex (SO), and nuclei of the lateral lemniscus (NLL). These hindbrain structures (NLL and SO) project to the inferior colliculus (IC, in midbrain) through both the excitatory and inhibitory inputs to IC. From IC, the information travels toward thalamus via relay in medial geniculate complex (MG), to reach the auditory cortex, associate auditory

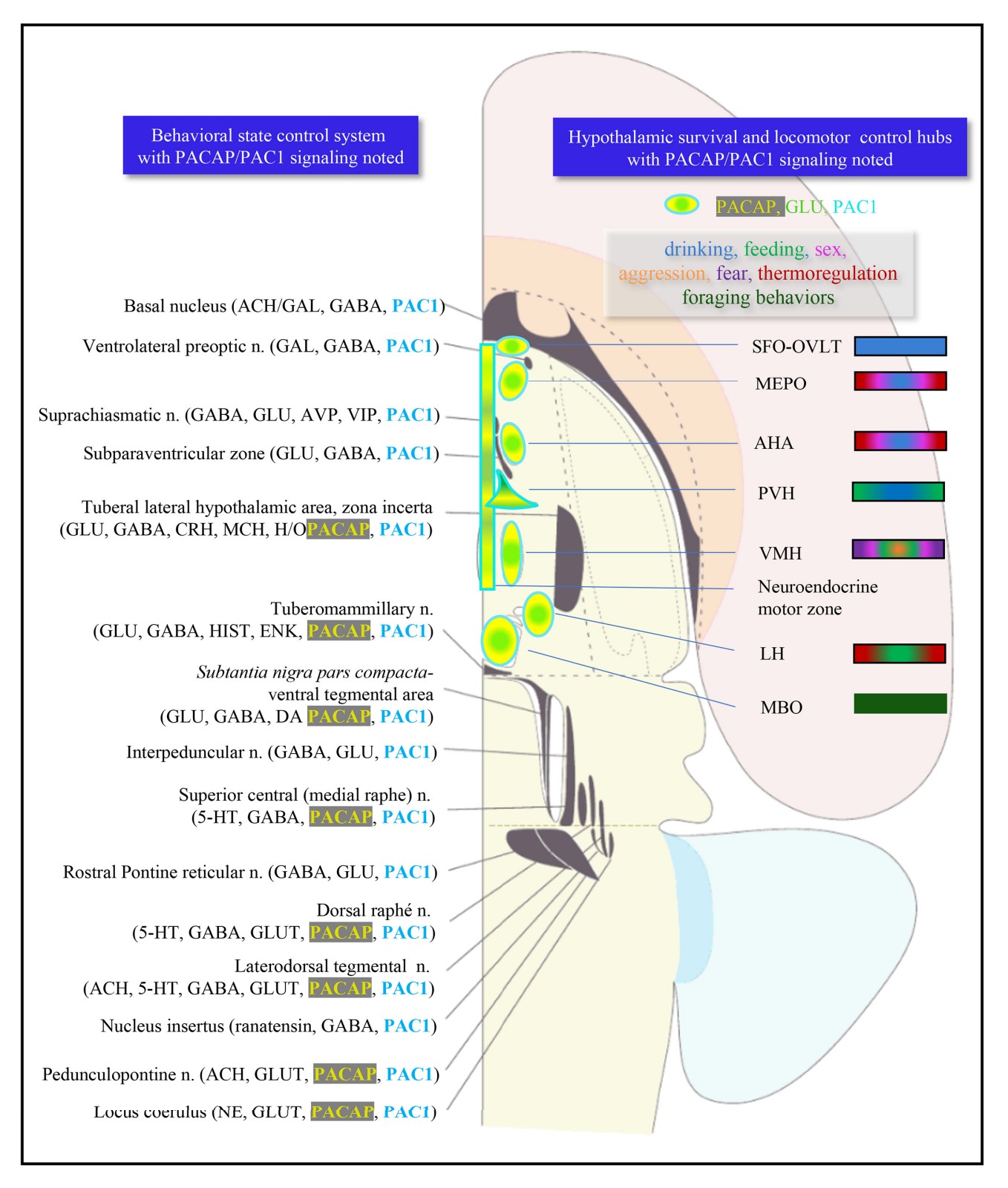

**Figure 7.** Presence of PACAP-PAC1 in major cells groups associated with behavioral state (left) and behavioral control system and hypothalamic instinctive survival system (right column). Left column: critical nodes for behavioral state symbolized by dark gray shaded objects, modified from *Swanson, 2012*. In the longitudinal cell group-column of brain stem the key neurotransmitters are annotated. ACH, acetylcholine; CRH, corticotropin-releasing hormone; DA, dopamine; ENK, encephalin: GABA, gamma-amino butyric acid; GAL, galanin; GLUT, glutamate; H/O, hypocretin/orexin; HIST,

*Figure 7 continued on next page*

*Figure 7 continued*

histamine; MCH, melanin-concentrating hormone; NE, norepinephrine; 5HT, serotonin. Right column: hypothalamic survival circuit that consisted of discrete hypothalamic regions contain interoceptors for a variety of substances and have neuronal afferences from primary sensory systems to control the secretory and instinctive motor outputs. The rectangle in the midline represents the neuroendocrine motor zone for secretion of hypophysiotropic hormones, which include thyrotropin-releasing hormone, corticotropin-releasing hormone, growth hormone-releasing hormone, somatostatin, gonadotropin-releasing hormone, dopamine, neurotensin. SFO, subfornical organ; OVLT, organum vasculosum of lamina terminalis; MnPO, preoptic nucleus; AHN, anterior hypothalamic area; PVH, paraventricular hypothalamic nucleus; VMH, ventromedial hypothalamic nucleus; LH, lateral hypothalamic area; MBO, mammillary body (for general reference see *Swanson, 2012*).

cortex, temporal association area; hippocampus, which subsequently projects to prefrontal cortices (*Ono and Ito, 2015*). Main neuronal structures that are relevant to the auditory pathway are shown in the schematic diagram of *Figure 6D*. The glutamatergic population of these structures were all found to co-express *Adcyap1*, *Slc17a6* and/or *Slc176a7*, as well as *Adcyap1r1* (*Figure 3A–F*).

The mainly *Sc321a1*-expressing structures located in subcortical regions which strongly co-expressed *Adcyap1r1* (*Figure 2D–G* and *Table 2*), include, for instance, BST and the CeA with its three subdivisions (*Figure 6D*, structures symbolized in pink-GABAergic, and blue outline–PAC expressing) participate in cognitive and emotional auditory information, obtaining auditory sensory input from brain stem>midbrain>thalamic MG, which project to lateral amygdala (LA, a main PACAP-containing nucleus derived from cortical plate) which send input to the cognitive centers through basal lateral amygdala (BLA, *Figure 2F and G*).

## Gustatory pathway

Chemicals (taste substances) in foods detected by sensory cells in taste buds distributed in the oro-pharyngeal epithelia are recognized as tastes in the gustatory cortex (GC), including granular insular cortex (GI) and agranular insular cortex (AI), also called dysgranular insular cortex (DIC), which is the primary gustatory cortex (*Figure 6E*). Between the peripheral sensory tissue and the GC, many neuronal hubs participate in the gustatory information processing, beginning with the geniculate ganglion (GG) of the facial nerve, cranial nerve CN-VII which receive taste stimuli from the anterior 2/3 of the tongue; the petrosal ganglion (PG) of the glossopharyngeal nerve, CN-XI, which receives the taste stimuli from the posterior third of the tongue; and nodose ganglion (NG), also called the inferior ganglion of the vagus nerve (CN-X, which innervates the taste buds of the epiglottis). The lingual nerve, a branch of mandibular nerve, from trigeminal nerve (CN-V), also innervates the anterior 2/3 portion of tongue. All these ganglia express PACAP as their co-transmitter and this expression is potentiated during injury (*Sundler et al., 1996*). Taste information from sensory ganglia of CN V, VII, IX, and X project to the rostral NTS of the medulla, which intensely expressed *Adcyap1* and *Adcyap1r1*. NTS neurons send taste information to parabrachial nucleus (PB, which also intensely expressed *Adcyap1* and *Adcyap1r1*) and then to the parvocellular division of the ventral posteromedial nucleus VPM of the thalamus, which projects to GC, as well as to the cognitive centers, central amygdala (CEA), bed nucleus of stria terminalis (BNST) and lateral hypothalamus (LH) which also reciprocally innervate the PB (*Halsell, 1992*). In *Figure 6* panel E, the basic wiring of taste circuit was based on the literature (*Halsell, 1992*; *Carleton et al., 2010*), and modified with PACAP-PAC1 signaling annotated (*Figure 1A–F*).

## PACAP-PAC1 signaling in major cell groups associated with behavioral state control system and hypothalamic instinctive survival system

In the course of a day, brain states fluctuate, from conscious awake information-acquiring states to sleep states, during which previously acquired information is further processed and stored as memories (*Tukker et al., 2020*). Anatomically and chemically distinct neuronal cell groups stretching from medulla, pons, midbrain, through the hypothalamus to the cerebral nuclei all participate in modulation of behavioral state (*Swanson, 2012*). We have briefly referred to sleep and waking behavioral states in analysis of the hypothalamic SCH in the visual/photoceptive pathways that project to midbrain and hindbrain behavioral state structures broadly defined. In *Figure 7*, left column we complemented the model of behavioral state cell groups and chemical signatures previously presented (*Swanson, 2012*), adding PACAP-PAC1 as informed by the current study (*Figure 1A–F*). Three behavioral state-related glutamatergic structures in medulla, that is *raphé magnus*, *raphé pallidus*,

and *raphé obscurus* are not represented in the left column in recognition of the original figure design of the author (Figure 9. 5, *Swanson, 2012*).

In the right column of *Figure 7*, we present the main hypothalamic survival and locomotor control hubs with PACAP/PAC1 signaling (*Figure 1A–F*) noted. These hubs are SFO, OVLT, MEPO, AHN, PVN, VMH, LH, MBO, and periventricular neuroendocrine motor zone. Recent evidence suggests that these cell groups control the expression of motivated or goal-oriented behaviors, such as drinking, feeding, sex, aggression, fear, foraging behaviors (*Figure 7*, color-filled rectangles represent the correlative behaviors with anatomical structure based on the literature [*Sternson, 2013*]). The rostral segment of this behavior control column has controllers for the basic classes of goal-oriented ingestive, reproductive and defensive behaviors, common to all animals, where the caudal segment has the controllers for exploratory behavior used to obtain any goal object (*Swanson, 2012*).

Other relevant structures more caudal in this longitudinal brain stem column are the reticular part of SN (not labeled in *Figure 7*), which is involved in the control of orienting movements of the eyes and head via projecting to SC, the VTA, *Adcyap1*-expressing, which together with ACB (*intensely Acyap1r1*-expressing) and STN (*intensely Adcyap1*-expressing), form the hypothalamic locomotor region.

After analyzing the above presence of PACAP and its receptor PAC1 in sensory-cognitive to motor output, a question emerged: what would be the behavioral and neuronal activation consequences if PACAP signaling were deficient?

## PACAP knockout impairs predator odor salience processing via reducing neuronal activation and vesicular transporter expression in key PACAP-PAC1 nuclei

To assess the behavioral implications of PACAP action within one of these circuits, we examined the effect of knockout of PACAP expression in brain on defensive behavior initiated via olfaction, using a predator odor paradigm (*Figure 8A and B*). Using a modified open-field box with a lidded container with cat urine litter, and wild type (WT) and PACAP-deficient (KO) C57Bl/6N mice (*Hamelink et al., 2002*), we assessed the defensive locomotion patterns during cat odor exposure, which include displacement pattern in the four quadrants of the box; approach to the stimulus (urine-containing cat litter) with complete retreats and incomplete retreats; and periods of immobility (freezing behavior). DISH for fos, *Slc17a7, Slc17a6*, and *Slc32a1* expression in the two experimental groups was performed and analyzed *post hoc*.

Cat urine triggered purposeful movement such as sniff>retreat in WT subjects (*Figure 8Ai*, green dashed lines indicate complete sniff>retreat cycles). In contrast, the movement patterns exhibited by KO subjects consisted in sniff>retreat cycles of short distances and the mice often returned to the container location area (*Figure 8Bi*, red dashed lines indicate incomplete/short sniff>retreat cycles). PACAP-deficient mice had significant reduction of complete retreats (p<0.001) and significant augmentation of incomplete retreats (p<0.001) (*Figure 8D*). *Figure 8Aii and Bii* show the X-Y dimension movement heat-maps for each group. Movement analysis by quadrants showed a significant increase in pixel values of the KO group in the quadrant where the odorant container was located (*Figure 8C*, p<0.05, C1). It is worth mentioning that movement in Z dimension (repetitive jumping), characteristic of PACAP-deficient mice (*Hashimoto et al., 2001*) was not represented in the X-Y dimension heat maps, and investigating the phenomenon is beyond the scope of the study. KO mice also exhibited significant reduction in freezing behavior (*Figure 8D*, p<0.001). To test whether or not the behavior of PACAP-deficient mice could be explained in whole or part by hypo- or anosmia, a standard test for olfaction was carried out. Wild-type and PACAPko mice were tested for the ability to retrieve a buried cookie as described in Materials and methods. Retrieval times for wild-type and PACAP-deficient mice were not significantly different (wild-type retrieval latency 38.0 ± 10.4 s; PACAPko retrieval latency 44.5 ± 12 s, p=0.7058, t-test, n = 6/group, each group comprising three male and three female mice approximately 8 weeks of age). We conclude that PACAP-deficient and wild-type C57Bl6/N mice were equally able to locate a food source using olfaction.

The *Fos* expression associated with the predator odor exposure behavioral test was significantly reduced, in PACAP-deficient mice, in olfactory areas, the mitral layer and glomerular layers, as well as higher predator-odor processing centers, for instance, the amygdaloid complex MEA and CEA, the LS, the endbrain ACA, PL and IL cortical areas, the BSTov; the hindbrain PB complex, which

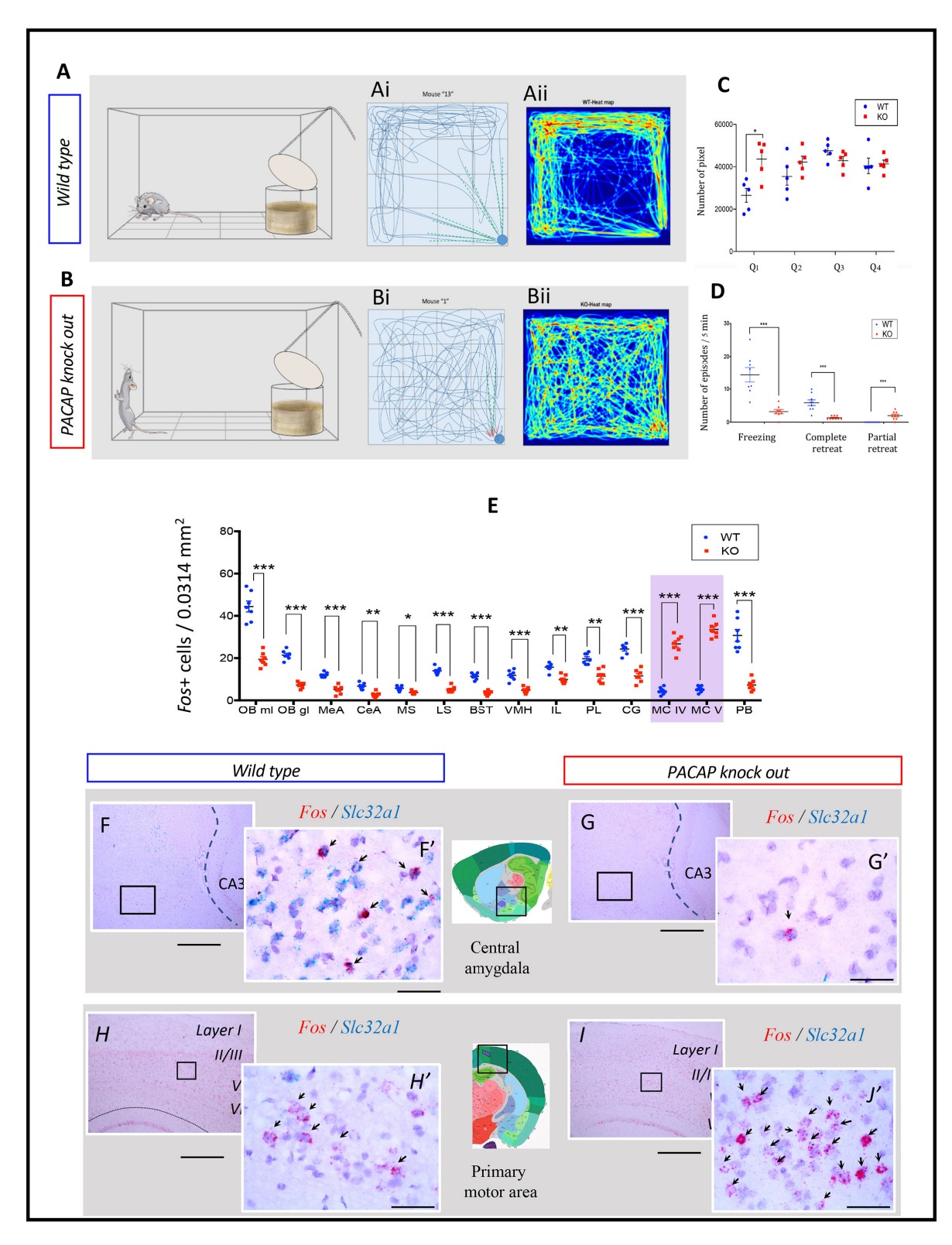

**Figure 8.** PACAP-deficient mice exhibited aberrant predator odor response and defensive behavior. (**A** and **B**) Schematics of behavioral procedure using a modified open field test (OFT) to study defensive behavior during predator odor exposure, with typical behaviors of wild type (A: freezing) and PACAP KO (B, wandering, hyperactive and jumping) symbolized. (**Ai** and **Bi**) representative 2D movement tracking in the OFT of WT (**Ai**) and OK (**Bi**) mice, with dashed lines symbolizing complete (green lines) /incomplete (red lines) approach/sniff > retreat cycles. (**Aii** and **Bii**) 2D movement heat-map, *Figure 8 continued on next page*

*Figure 8 continued*

WT, *n* = 9 mice; KO, *n* = 9 mice. (C) Pixel analysis reflecting total distance traveled in each of the four quadrants, being the Q1 the container located quadrant (clockwise numbering). The pixel numbers of per quadrant (Q) per mouse were compared that revealed a significant increase only in Q1 (WT vs KO, 26500 ± 3288 vs. 43640 ± 3921, *p<0.05. (D) Numbers of freezing, complete sniff>retreat or partial sniff>retreat episodes were assessed using Student t-test. For freezing (n = 8) and complete sniff>retreat (n = 9) behaviors, KO mice showed significant reduction WT:14.29 ± 2.18 vs KO: 3.18 ± 0.64; and WT: 5.89 ± 0.84 vs KO: 1.33 ± 0.17; for partial sniff>retreat behavior (n = 9) KO mice showed significant increase (WT: 0 vs KO: 2 ± 0.41). (E) Number of cells expressing *fos* mRNA 45 min after the predator odor test were quantified in a 0.0314 mm$^2$ area and statistic significant differences between KO and WT were determined using a multiple t-test and the Bonferroni-Dunn method. There was a significant decrease in the expression of *fos* in the following regions of KO mice: olfactory bulb mitral layer 'OB ml' (WT: 44.4 ± 2.60 vs KO: 19.43 ± 1.25); olfactory bulb granule layer 'OB gl'(WT: 21.43 ± 0.84 vs KO: 7.29 ± 0.52); medial amygdala 'MeA' (WT: 12 ± 0.44 vs KO: 5 ± 0.76); central amygdala 'CeA' (WT: 6.71 ± 0.57 vs KO: 2.71 ± 0.47); medial septum 'MS' (WT: 5.71 ± 0.42 vs KO: 3.71 ± 0.29); lateral septum 'LS' (WT: 14 ± 0.62 vs KO: 5.14 ± 0.56); bed nucleus of stria terminalis 'BST' (WT: 11.43 ± 0.57 vs KO: 3.57 ± 0.37); ventromedial hypothalamus 'VMH' (WT: 11.86 ± 0.91 vs KO: 4.86 ± 0.50); infralimbic cortex 'IL' (WT: 15.71 ± 0.714 vs KO: 9.86 ± 0.74); prelimbic cortex 'PL' (WT: 19.71 ± 0.97 vs KO: 11.43 ± 1.23); cingulate cortex 'CG' (WT: 24.28 ± 0.92 vs KO: 11.57 ± 1.15) and parabrachial nucleus 'PB' (WT: 30.71 ± 2.82 vs KO: 7.29 ± 1.02) and granule cell layer of cerebellum 'CBgcl' (WT: 13.14 ± 0.96 vs KO: 8 ± 0.76). In contrast, the number of *fos* nuclei was augmented in two motor related areas in KO animals related to WT: motor cortex, layer IV 'MC IV' (WT: 4.43 ± 0.65 vs KO: 26.71 ± 1.54) and motor cortex, layer V (WT: 5.28 ± 0.57 vs KO: 33.57 ± 1.43). (F and G) example of reduced *fos* expression pattern in central amygdala in KO mice and (H and I) augmented *fos* expression in primary motor cortex of KO mice. Scale bars: 200 µm for low-magnification and 20 µm for high-magnification insets. Statistic significant differences are depicted as ***p<0.001, **p<0.01, *p<0.05. Note that during the freezing analysis, two outliers, one in each experimental group were discarded because being 45 and 24 (counts), with 5 times and 11 times higher than the standard deviations for the corresponding groups, that is WT:14.29 ± 6.17 (mean ± SD) vs KO: 3.18 ± 1.80 (mean ± SD). This exclusion was based on criteria explained in NIH Rigor and Reproducibility Training course, https://www.nigms.nih.gov/training/documents/module4-sample-size-outliers-exclusion-criteria.pdf and http://www.itl.nist.gov/div898/handbook/prc/section1/prc16.htm.

The online version of this article includes the following source data for figure 8:

**Source data 1.** MatLab script for motion heatmap analysis.
**Source data 2.** Raw data for panels C, D and E.

provides the main upstream PACAP signaling source especially for BSTov and CEA (http://connectivity.brain-map.org/) for the above cognitive centers, and the hypothalamic VMH, which control the aggressive behavior (*Figure 8E–G*), as well as the cerebellar cortex in which we found reduced *fos* expression in the granule cell layer of the ansiform lobule, which is reported to influence limb movement control (*Manni and Petrosini, 2004*; *Zhu et al., 2006*). In contrast, KO mice had increased *Fos* expression, compared to wild type, in the motor cortex layer VI and V (*Figure 8E,H and I*).

We assessed *fos* expression in the PACAP/PAC1 system including both glutamatergic and GABAergic neurons in response to odorant exposure, using DISH technique and *Slc17a7*, Slc17a6, and Slc32a1 probes to identify the glutamatergic and GABAergic neurons. Surprisingly, we observed a sharp reduction in the abundance of three vesicular transporters in the main PACAP-containing nuclei we described vide supra (*Figure 9*). This reduction was observed both as reduced abundance of the ISH staining puncta at the single-cell level and the density of expressing cells in the given region (*Figure 9E*).

## Discussion

Here, we describe in detail the overall topographical organization of expression of mRNA encoding PACAP, and that of its predominant receptor PAC1, and their co-expression with the small-molecule transmitters, glutamate and GABA, using probes for mRNA encoding the vesicular transporters VGLUT1/2 and VGAT, as obligate markers for glutamatergic and GABAergic phenotypes, respectively, in mouse brain. A highly sensitive dual in situ hybridization method (RNAscope 2.5HD duplex detection) was used and corroborated with Allen Brain Atlas ISH data throughout brain. We report glutamatergic or GABAergic identity of 181 *Adcyap1*-expressing cell groups as indicated by co-expression of corresponding vesicular transporters. All *Adcyap1*-expressing neurons studied, and most of their neighboring cells, co-express mRNA encoding PAC1 (*Adcyap1r1*), indicating that the PACAP/PAC1 pathway, besides using classical neurotransmission through axon innervation and transmitter release, likely employs autocrine and paracrine mechanisms for signal transduction as well.

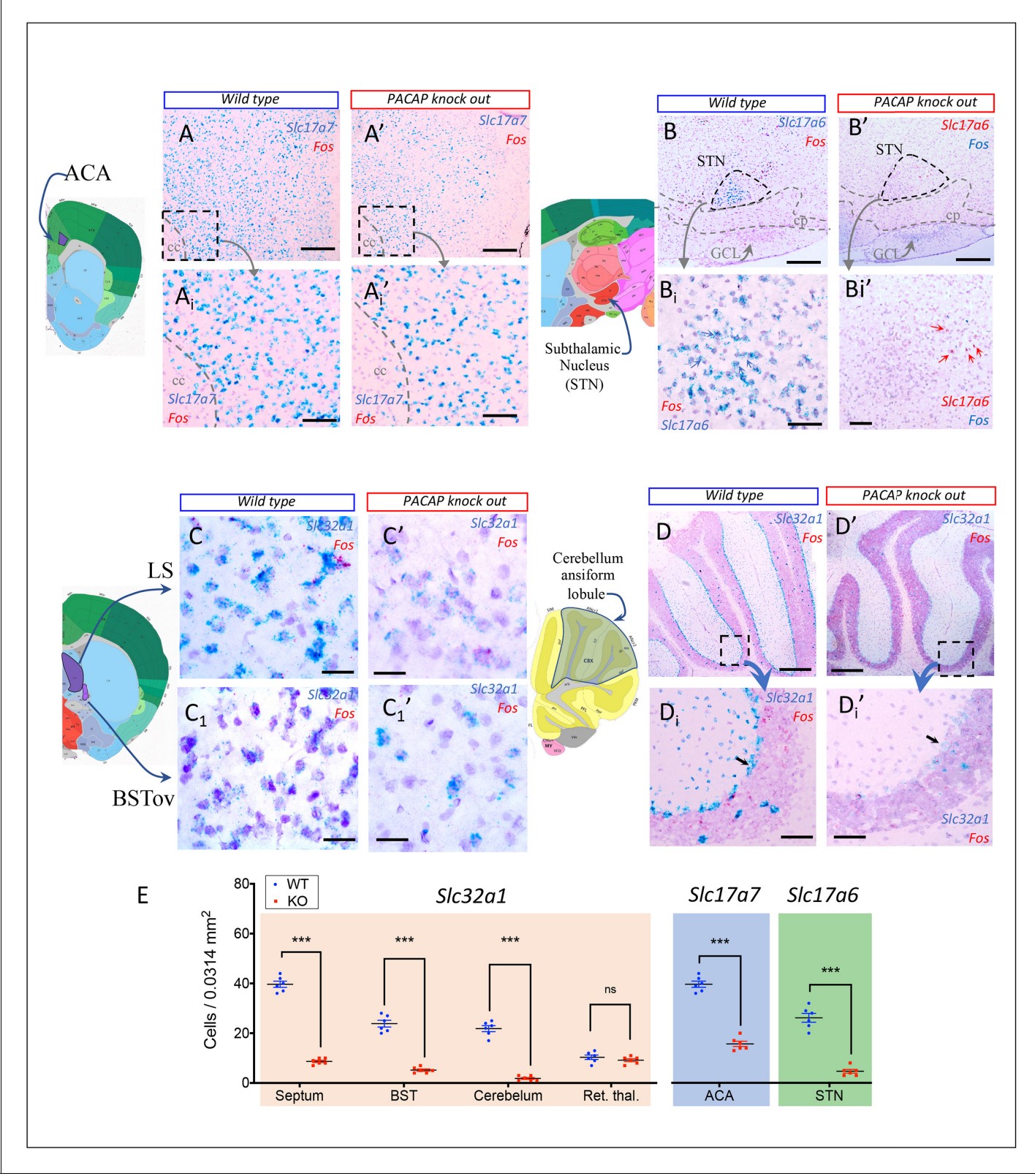

**Figure 9.** PACAP-deficient (KO) mice showed significant down-regulation of vesicular transporters for glutamate and GABA in regions where *Adcyap1* or *Adcyap1r1* were strongly expressed in WT mice. (A–D) Examples of in situ hybridization using RNAscope method showing down-regulation in KO mice (A', B', C', D') of *Slc17a7* (VGLUT1 mRNA) in anterior cingulate area (ACA) (panels As), *Slc17a6* (VGLUT2 mRNA) in hypothalamic subtalamic nucleus (STN) (Panels B) and of *Slc32a1* (VGAT mRNA) in lateral septum (LS) and bed nucleus of stria terminalis, oval subnucleus (BSTov) (panels Cs),
*Figure 9 continued on next page*

*Figure 9 continued*

and cerebellar cortex, the ansiform lobule's, Purkinje's cells (panels Ds). Note that the feature of reduced expression of *Slc32a1* at both single-cell and cell density levels, can be clearly observed (arrows) in the Purkinje cells. (E) Number of cells expressing *Slc32a1* (orange shading), *Slc17a7* (blue shading) and *Slc17a6* (green shading) were quantified in a 0.0314 mm$^2$ area and statistically significant reductions of KO compared to WT were detected septum (WT: 39.67 ± 1.23 vs KO: 8.67 ± 0.49); BST (WT = 23.83 ± 1.35 vs KO = 5.17 ± 0.48); ansiform lobule of cerebellum (WT: 21.83 ± 1.19 vs KO: 1.83 ± 0.4); anterior cingulate area (WT: 39.67 ± 1.23 vs KO: 15.67 ± 1.05) and subthalamic nucleus (WT: 26.17 ± 1.78 vs KO: 4.67 ± 0.76), while in the reticular thalamic nucleus, a region that did not contain *Adcyap1* expressing cells, no significant difference was detected (negative control). Statistic differences are depicted as ***p<*0.001 and* ns: not significant. Scale bars: A and A': 300 μm; Ai and Ai': 100 μm; Bs and Cs: 20 μm; D and D': 500 μm; D1 and D1': 100 μm.

The online version of this article includes the following source data for figure 9:

**Source data 1.** Raw data for panel E.

With the information obtained by PACAP/PAC1 expression and co-localization with small-molecule transmitters, we explored the hypothesis that the broad brain distribution of PACAP may reflect a more specific physiological function at a systems level. We examined the distribution of PACAP/ PAC1 expression within specific sensory input-to-motor output pathways passing through the cognitive centers mentioned in these studies, in which most of the hubs analyzed used PACAP>PAC1 signaling within the context of glutamate/GABA neurotransmission. This systematic analysis has revealed several possible PACAP-dependent networks involved in the highest levels of motor control,that is the hypothalamic pattern initiator and controller system, intermediate (or intervening) levels of somatosensory information processing, and the complex cognitive and behavioral state control for behavior.

## Newly identified PACAP-expressing cell groups in mouse brain suggesting circuit-level function

The high-resolution DISH experiment and analysis showed that although the hypothalamus has the highest group-density of *Adcyap1*-expressing subpopulations in the brain, PACAP should not be considered as mainly a hypothalamic neuropeptide. Our data revealed newly identified *Adcyap1*-expressing neuronal populations with mainly *Slc17a7* co-expression, derived from both *cortical plate* and *brain stem*. Interestingly, most of these neurons co-expressed *Calb2*, the mRNA transcript encoding the calcium binding protein calretinin. The prominent regions/nuclei of this group include: (1) regions derived from the cortical plate, that is MOB, AOB, AON, TT, PIR, NLOT, COAa, related directly to olfactory processing, (2) regions derived from cortical subplate, that is hippocampus CA3vv subset of pyramidal neurons in the ventral pole, dentate gyrus mossy cells, and posterior amygdalar nucleus and bed nucleus of anterior commissure; (3) brain stem structures that is pontine grey, Koelliker-Fuse of parabrachial complex, nucleus of lateral lemniscus within pons; nucleus of tractus solitarius, dorsal and ventral vestibular nucleus, and superior olivary complex lateral part.

## Bed nucleus of anterior commissure (BAC): a prominent yet chemoanatomically ill-defined PACAP-expressing nucleus

We described this nucleus as a newly identified major *Adcyap1*-expressing nucleus (section Structures derived from cerebral nuclei; *Figure 5*). Regarding the phylogenetic classification of the BAC, some authors have argued that "the septum (within striatum) is divided into the lateral, medial, and posterior septum (LS, MS and PS, respectively); and the PS is further subdivided into the triangular septum (TS) and the bed nucleus of the anterior commissure (BAC)" (*Risold, 2004*). However, the Allen Brain Map classifies the BAC as a pallidum structure (https://portal.brain-map.org/). Although literature on BAC connectivity is sparse, LS is identified as one of the inputs to the BAC (*Swanson et al., 2016*) and the medial habenula is one of the reported target region, with implication in the control of anxiety and fear responses (*Yamaguchi et al., 2013*), the connectivity and function of this glutamatergic-PACAP nucleus is largely elusive. Hence, the chemical identification of this nucleus opens new opportunities for generation of animal models using optogenetic/chemogenetic tools to discern the role of this structure within behavioral circuit(s).

## A novel transcriptomically distinct pyramidal subpopulation in ventral hippocampal CA3c is well-placed for modulation of the predator threat response

The hippocampus is typically described in the context of the tri-synaptic circuit. The tri-synaptic circuit is composed of three sequential glutamatergic synapses: perforant path axons of layer II neurons in entorhinal cortex project to the outer two-thirds of the dentate gyrus molecular layer, the location of the distal granule cell dendrites; mossy fiber axons of granule cells project to proximal dendrites of area CA3 pyramidal cells; and the Schaffer collateral axons of CA3 pyramidal cells project to stratum radiatum of CA1, where the apical dendrites of area CA1 pyramidal cells are located (*Amaral and Witter, 1989*; *Figure 4E*). However, this trisynaptic circuit seems insufficient to explain the ventral hippocampus observations about the CA3c pyramidal *Slc17a7/Adcyap1* co-expressing neurons. Hence, this is a surprising finding since, as discussed by *Scharfman, 2007*, ventral CA3 may be a point of entry that receives information, which needs to be 'broadcast,' such as for stress responding, whereas the dentate gyrus may be a point of entry that receives information with more selective needs for hippocampal processing (*Scharfman, 2007*). It has been reported that the CA3c pyramidal cells possess collaterals that project in the opposite direction to the tri-synaptic circuit, 'back' to the dentate gyrus, by either direct innervation of the mossy cells and GABAergic interneurons in the hilus, or to the granule cell layer glutamatergic and GABAergic neurons (*Scharfman, 2007*). Those targeted cell types strongly expressed *Adcyap1r1* (*Figure 2A and A'*). A hypothetical circuit modified from the literature (*Scharfman, 2007*), including this newly identified subset of CA3c PACAP-expressing cells, is presented here (*Figure 4F*). The hippocampal formation has been suggested to act not as a unitary structure, but with the dorsal (septal pole, DH) and ventral (temporal pole, VH) portions performing different functions (*Moser and Moser, 1998*). Their argument is based on three data sets. First, prior anatomical studies indicated that the input and output connections of the dorsal hippocampus (DH) and ventral hippocampus (VH) are distinct (*Swanson and Cowan, 1977*). Second, spatial memory appears to depend on DH not VH (*Moser et al., 1995*). Third, VH lesions, but not DH lesions, alter stress responses and emotional behavior (*Henke, 1990*). Here, we contribute with a fourth element, which is the subset of CA3 pyramidal neurons, located in the temporo-ventral pole of the hippocampal formation, transcriptomically distinct from the rest of CA3 pyramidal neurons, based on Slc17a7, Adcyap1 and Calb2 expression, which latter two elements are absent in the septo-dorsal pole.

The VH has attracted attention in odor studies due to dense reciprocal connections to the MeA and to other amygdalar nuclei such as cortical nucleus that receives input directly from the main olfactory system (*Scalia and Winans, 1975*; *Figure 6*). The VH also project to AOB and the piriform cortex, a major target of the MOB (*Shipley and Adamek, 1984*). It was reported that rats with VH lesions exhibited deficits in freezing and crouching when exposed to cat odor (*Pentkowski et al., 2006*). Another study in mice exposed to coyote urine showed that VH lesions impaired avoidance and risk assessment behaviors (*Wang et al., 2013b*). In the experiments reported here, PACAP-deficient mice, with no neurons expressing *Adcyap1* in CA3vv of VH, yielded similar behaviors, suggesting this cell subpopulation may contribute to optimal predator odor processing using PACAP as co-transmitter to coordinate the predator threat response.

## The autocrine/paracrine and neuroendocrine nature of PACAP>PAC1 signaling and glutamate and GABA vesicular transporter expression

The analysis of our data showed an impressive amount of Adcyap1 and Adcyap1r1 coexpression at both single-cell and regional levels (*Figure 3*), suggesting that the PACAP/PAC1 pathway uses *autocrine* and *paracrine* mechanisms in addition to classical neurotransmission through axon innervation and transmitter co-release (*Hökfelt et al., 1984*; *Zhang and Eiden, 2019*). That is, besides sending PACAP-containing axons to innervate regions where PAC1 is strongly expressed, PACAP may be released through soma and dendrites, to bind PAC1 expressed on the same and neighboring cells, to prime the neuron for optimum function (*Leng, 2018*). We still know relatively little about the possible functional interactions and control of release of PACAP>PAC1 signaling in this aspect. However, the observation reported in this study about down-regulation of vesicular transporter mRNAs in PACAP knockout mice, in regions/cell populations which were normally *Adcyap1*-expressing in wild-type mice, provides one possible example of such a function for PACAP. Activity-dependent

regulation of both glutamate and GABA vesicular transporter synthesis and membrane insertion has been reported (*De Gois et al., 2005*; *Erickson et al., 2006*; *Doyle et al., 2010*). PACAP signaling most commonly leads to a net increase in neuronal excitability through modulation of intrinsic membrane currents and transiently increasing intracellular calcium concentration (*Johnson et al., 2019*). Decreased vesicular transporter mRNA expression would be expected at the cellular level to decrease transmitter quantal size, thus decreasing excitatory/inhibitory postsynaptic currents (EPSCs and IPSCs, respectively) (*Billups, 2005*), and is consistent with our behavior data. PACAP absence in KO mice resulted in a behavioral hypoarousal response to moderate predator-odor stimulus. The vesicular transporter mRNA loss observed at RNA level upon complete PACAP deficiency, revealed in predator odor-exposed mice, may reflect a modulatory role for PACAP, in wild-type animals, on vesicular transporter mRNA (and protein) abundance within a more restricted, but functionally relevant, range depending on the level of PACAP release and autocrine signaling.

PACAP-deficient mice show hyperlocomotion and abnormal gait, as well as bouts of repetitive jumping (data not shown, but see *Hashimoto et al., 2001*, *Gaszner et al., 2012*, *Hattori et al., 2012*). In 1930, Hinsey, Ranson and McNattin reported a visionary experimental result, done in cats and rabbits about the rôle of the hypothalamus in locomotion (*Hinsey, 1930*). Quoting Swanson's interpretation of this early experiment: "when the central nervous system is transected roughly between the mesencephalon and diencephalon, the animals displayed no spontaneous locomotor behavior. They remain immobile until stimulated. On the other hand, animals with transection roughly between diencephalon and telencephalon (or upon complete removal of the cerebral hemispheres, and the thalamus) display considerable spontaneous behavior. In fact, they can be *hyperactive* when the transection is a bit caudal, but cannot spontaneously eat, mate, or defend themselves. Notably, these last three functions are preserved at a primitive level when the transection is just slightly more rostral". This evidence, combined with the selective lesions or stimulations of the hypothalamus, suggests that the ventral half of the diencephalon (hypothalamus) contains neural mechanisms regulating setpoints for locomotor and other classes of motivated behavior' (Figure 8.9, *Swanson, 2012*). As reported here, the major cell groups in the ventral diencephalon, associated with behavioral state (*Figure 7*, left column) and behavioral control (*Figure 7*, right column) all use PACAP/PAC1 signaling for co-transmission. Hence, movement pattern/initiation disorders would be expected in PACAP-deficient animals. Moreover, we found marked reductions in *Slc32a1* in Purkinje cells (PC) of the ansiform lobule of the cerebellum, as well as significant reduction in *Fos* expression in the granule cells (*Figure 9* panels Ds), the main excitatory input of the PC (*Manni and Petrosini, 2004*). The reduced VGAT availability in PC in the lateral lobules of the cerebellum, involved mainly in coordinating the movement patterns of the limbs (*Manni and Petrosini, 2004*, *Sakayori et al., 2019*), necessarily weakens the inhibitory influence on the excitatory output of the deep cerebellar nuclei which make ascending projections to several forebrain and midbrain structures, including hypothalamus (*Zhu et al., 2006*). In *Figure 10*, we propose a model based on the analysis of this study to provide a rationale for hyperlocomotion and hypoarousal behavior observed in PACAP-deficient mice. Under normal conditions, the hypothalamic motor initiator/controller centers (*Figure 7*, longitudinal PACAP cell groups in hypothalamus) are modulated, for instance, by direct excitatory (e.g. from deep cerebellar nuclei, DCN), indirect excitatory (through disinhibition of GABAergic projections from BNST, CEA to GABAergic neuronal circuits of hypothalamus) and inhibitory (from lateral septum) influences (*Swanson, 2012*; *Figure 10A*). We showed in this study that PACAP deficiency caused sharp downregulation of both vesicular glutamate and GABA transporters in the key nuclei that in wild-type mice Adcyap1 is expressed, such as cerebellar Purkinje cells that send inhibitory input to DCN; the prefrontal cortex, that with reduced VGLUT1 expression (*Figure 9*), could result in a deficient activation of striatum and pallidum cognitive structures, that send GABAergic input to the hypothalamic centers, through direct inhibition or indirect excitation through disinhibition mechanisms, for locomotor patterning and initiation. Unbalanced excitation/inhibition occurring in this hypothalamic region may result in hyperexcitation to motor pathways ending in primary motor cortex (*Figure 8*, higher fos expression in primary motor cortex in KO subjects), which command the spinal motor neurons for muscle activities (*Figure 10B*).

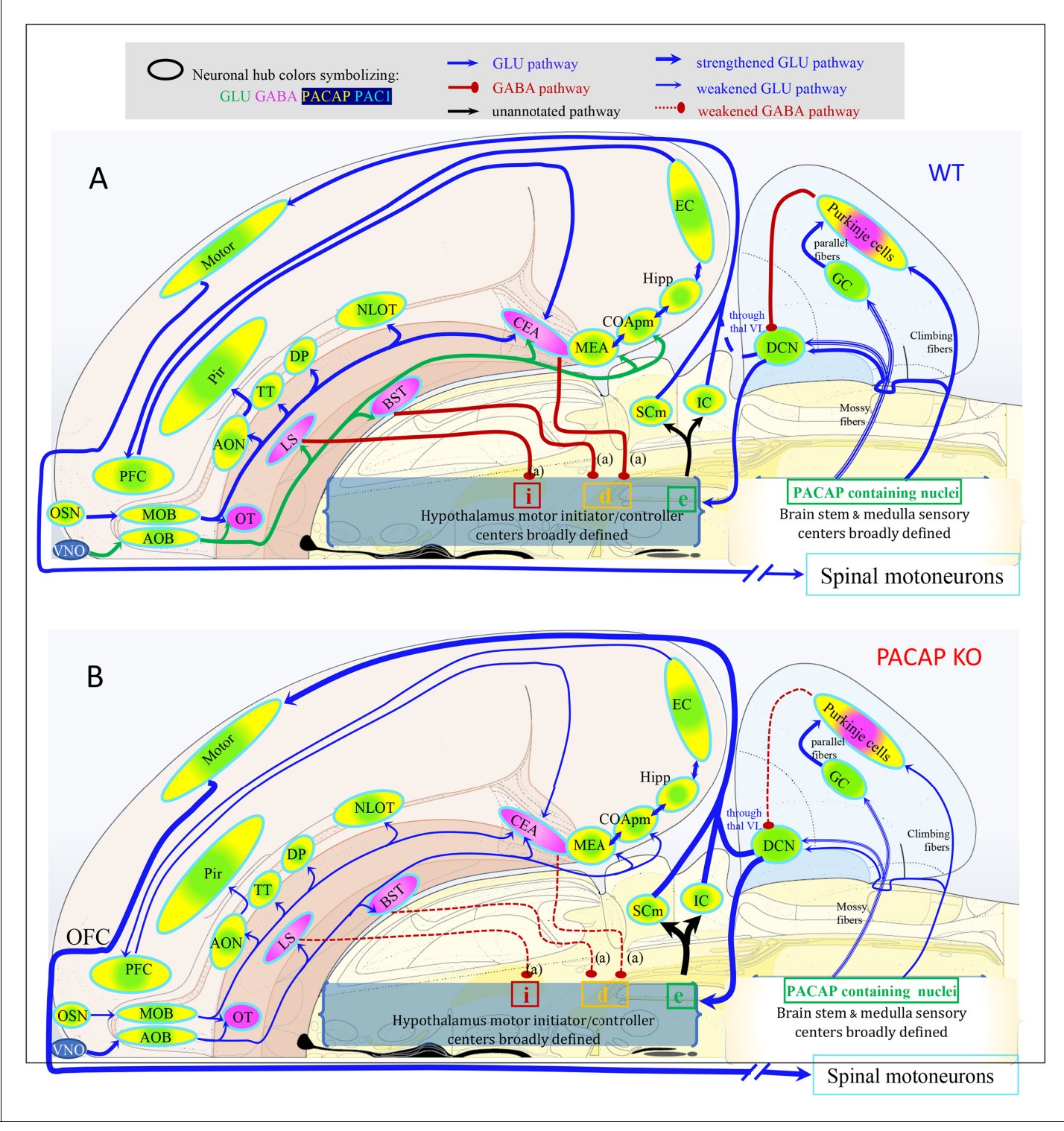

**Figure 10.** Proposed model to interpret how PACAP deficiency influence the olfactory information salience processing to impact motor output, bases on the analysis of this study. (**A**) Schematic representations of the mouse peripheral and central olfactory pathways to cognitive centers and motor higher control centers in brain stem under normal condition. PACAP-Pac1 glutamatergic/GABAergic signaling is symbolized by colors. (**B**) PACAP deficiency weakens the corresponding neuronal hubs inducing ultimately unbalanced excitation/inhibition in sensory-cognitive and motor pattern initiator and controller centers, which may underlie the hyperactivity and attention deficit to salient olfactory stimulus during predator odor exposure test. OSN: olfactory sensorial neurons; VNO: vomeronasal organ; MOB: main olfactory bulb; AOB: accessory olfactory bulb; AON, anterior olfactory n.; TT: taenia tecta; OT: olfactory tubercle; DP: dorsal peduncular area; Pir: piriform cortex; NLOT: n. lateral olfactory tract; EC: entorhinal cortex; PFC:

*Figure 10 continued on next page*

prefrontal cortex; COApm: corticoamygdalar, posteromedial; MEA and CEA: medial and central amygdala; LS: lateral septum; BNST: bed nucleus of stria terminalis; SCm: superior colliculus, motor related; IC: inferior colliculus; DCN: deep cerebellar nuclei; GC: granule cells; thal VL: thalamic ventrolateral n.; i: inhibition; d: disinhibition; e: excitation.

# Materials and methods

## Key resources table

| Reagent type (species) or resource | Designation | Source or reference | Identifiers | Additional information |
|---|---|---|---|---|
| Strain, strain background (*Mus musculus*), both sexes | Wildtype C57Bl/6N | In-house breeding program in accordance with NIH guidelines and standards and housed in cages containing 3–5 male siblings. | C57Bl/6N | |
| Strain, strain background (*Mus musculus*), both sexes | PACAP KO mice C57Bl/6N | Hamelink et al Proc Natl Acad Sci, 2002, 99(1):461–6, PMID:11756684 NIH in-house breeding program | C57Bl/6N Adcyap1-/- | |
| Commercial assay or kit | RNAscope detection assay probes | 2020 Advanced Cell Diagnostics, Inc | 405911 409561 319171 416331 319191 316921 502231 465391 404631 421931 316091 | *Adcyap1* *Adcyap1r1* *Slc17a6* *Slc17a7* *Slc32a1* *Fos* *Vipr1* *Vipr2* *Sst* *Pvalb* *Crh* |
| Commercial assay or kit | RNAscope 2.5 HD Duplex detection kit Detection reagent-red | 2020 Advanced Cell Diagnostics, Inc | —————— 322500 322360 | |
| Software, algorithm | MATLAB scripts | This paper | Script | For motion heatmaps |

## Mice

The following male mouse strain/line were used in this study: C57BL/6N and C57BL/6N PACAP deficient, age 8–10 weeks, generated through an in-house breeding program in accordance with NIH guidelines and standards. Generation of PACAP knock-out mice have been described previously (*Hamelink et al., 2002*). All experiments were approved by the NIMH Institutional Animal Care and Use Committee (ACUC) and conducted in accordance with the NIH guidelines.

## In situ hybridization (ISH): RNAscope single and duplex (DISH) procedure for PACAP mRNA co-distribution mapping

Mice (n = 5) were deeply anesthetized with isoflurane, decapitated and whole brains were removed, embedded in OCT medium and rapidly frozen on dry ice. Serial brain sections (12 μm thick) were cut on a cryostat (Leica CM 1520), adhered to SuperFrost Plus slides (ThermoScientific). RNA probes were designed and provided by Advanced Cell Diagnostics (Hayward, CA): Mn-Adcyap1 (gene encoding PACAP), Mn-Adcyapr1r (gene encoding PAC1), Mn-Slc17a7 (gene encoding VGLUT1), Mn-Slc17a6 (gene encoding VGLUT2), Mn-Slc32a1 (gene encoding VIAAT, also called VGAT), Mn-Vipr1 (gene encoding VPAC1), Mn-Vipr2 (gene encoding VPAC2), Mn-Sst (gene encoding somatostatin), Mn-Pvalb (gene encoding palvalbumin), Mn-Crh (gene encoding corticotropin releasing hormone), and Mn-Fos (gene encoding cFos). All staining steps were performed following the protocols provided by the manufacturer for chromogenic detection of mRNA in fresh frozen tissue samples. Stained slides were examined with both light microscope with digital camera and were image-captured with 20X objective with ZEISS Axio Scan (Carl Zeiss Microscopy, Thornwood, NY from Systems

Neuroscience Imaging Resource, NIMH-IRP, NIH) and images for sections from each animal were organized and converted to TIF files with BrainMaker (MBF Bioscience, Williston, VT).

## Chemo-anatomical analysis and circuit identification

Anatomical nomenclature and regional delineation were done according to Allen Mouse Brain Atlas (http://www.allenbrainatlas.org) corroborated with Paxinos and Franklin's the Mouse Brain in Stereotaxic Coordinates. The *Adcyap1*/*Adcyap1r1* distribution mapping and semi-quantification and co-expression with glutamate/GABA transporter mRNAs were corroborated with the Allen brain atlas https://mouse.brain-map.org/. For semiquantitative scoring we used criteria similar to previous literature reports (*Hannibal, 2002*). Our annotation criteria were: the number of expressing cells as a percentage of total Nissl stained nuclei: '-', not observed; '+', weak (<20%); '++', low (20–40%); '+++', moderate (40–60%); '++++', intense (80–60%); '+++++', very intense (>80%). Functional neuroanatomy order and annotations are based on Allen Institute Mouse Reference Atlas (http://atlas.brain-map.org/). For schematic circuit chartings of *Figures 6*, *7* and *10* we used Swanson's rodent flat maps downloaded from http://larrywswanson.com/?page_id=164. Sensorimotor circuit hubs and connectivity of *Figures 6* and *10* are based on literature cited in the results section as well as consulting the website https://sites.google.com/view/the-neurome-project/connections/cerebral-nuclei?authuser=0.

## Behavioral experiments: predator odor

### Subjects

C57BL/6N wild-type and PACAP-deficient male mice, age 8–10 weeks, n = 9 (N = 18) were used. They were group-housed (4/cage) in a room kept on a controlled light-dark cycle (light on 7:00 am and off 7:00 pm) under constant humidity and temperature conditions.

### Odor stimuli

Cat urine material was collected from domestic cat litter, where multiple male and female cats used the same box to urinate.

### Modified open-field box for odor exposure and fos expression assessment

A custom-made wooden box (28 x 28 x 28 cm, with a sliding glass lid) was used (see *Figure 8A*). The box was positioned inside a low-noise suction hood, then in a lidded container, cat urine containing litter material was introduced into the box (*Figure 8A*). After introducing the subject, the lid of container was opened through a string and the box was closed to avoid the odor to escape. Video recording of the animal behavior was made during a 10-min period, after which animals were returned to its home cage for 30 min and then euthanized by cervical dislocation and brains rapidly removed and processed for dual in situ hybridization (DISH) using the RNAscope 2.5 HD Duplex Assay to assess the expression of *fos* mRNA within *Adcyap1*-, *Adcyap1r1*-, *Slc17a7*-, *Slc17a6*-, and *Slc32a1*-positive cells.

### Behavior assessment

Mouse behavior during the 10 min of cat urine exposure was recorded by an overhead camera. Behavioral scoring was performed off-line with *blind* analysis (performed by ECG, medical student in laboratory rotation, see acknowledgement). Freezing behavior was assessed in the first 5 min lapse, giving a score every 5 s when the subject exhibited immobility with piloerection. Complete cycle of sniff>retreat was defined as the movement of approach the containing and immediately retreat beyond the quadrant I (QI, *Figure 8*) while incomplete sniff > retreat cycles were defined when the subject approached the container and stayed in the proximity or retreat only a short distance within the same quadrant (Q1). Mice spatial displacement maps were produced on-screen by the analyst with aid of the software PowerPoint > shape format > curve tool (Microsoft Office). The experimental arena was represented in a slide with a square containing 16 small squares. The flask containing the cat litter is symbolized with a filled circle. The computer aided manual drawing was visually guided by the video recording under the strict criterion that the analyst makes a click in the corresponding position of the map for each *end of a lineal movement* of the mouse. The jumping behavior is only symbolized with 'zig-zags'. Four of mice exhibited this behavior and due to the aim of this

study – the place preference assessment – these subjects were discarded for the Heat map. Power-Point traces where skeletonized with ImageJ and movement traces from n = 5 animals were used. The 2D movement heat-map was produced with MATLAB R2016b, for each time the mouse passed through a point in space, the value of 1 was assigned. Using the MATLAB 'color dispersion function' (PSF) and 'color map' functions, a heat map was constructed. The trajectories of the five mice in each group were superimposed, then using the 'bwarea' function, a count of the pixel quantity per quadrant was performed, being Q1 the container located quadrant (clockwise numbering). Thereby, each pixel in the image had a different value and color according to the number of times the mouse was at that point. The number of pixels per quadrant for mice in the KO group was compared with mice in the WT group following one-way ANOVA.

### Behavioral experiments: olfaction

A standard test for olfaction as described by Yang and Crawley, Current Protocols in Neuroscience 8.24.1–8.24-12, July 2009 was carried out in wild-type and PACAP-deficient mice. Briefly, mice were individually housed under standard light-dark conditions for a period of four days. Subsequently, a cookie (Chocolate Teddy Grahams—Nabisco) was placed in each cage at mid-day on two successive days, consumption was confirmed, and all food was removed on the afternoon of the second day. The following day, mice were acclimated to the testing room for 1 hr, and testing initiated by placing mice in a fresh cage (test cage) with bedding 3 cm deep, allowing exploration for 5 min; removal to a fresh cage while a cookie was buried 1 cm deep at one end of the test cage, and placement at the opposite end to the buried cookie in the test cage. Time for mouse to retrieve the buried cookie and hold the cookie in its forepaws and begin to eat was recorded.

### Assessment for *Fos* and vesicular transporter expression

The counting for *Fos* and vesicular transporters expressed in cells was done on a computer screen connected to a digital camera mounted over a light microscope. The region of interest (ROI) was centered through observation using the microscope oculars and 20x objective and projected to a large computer screen through a digital camera. A fixed square equivalent to 0.0314 $mm^2$ for the magnification/computer enlargement was pre-fixed and moved to the ROI choosing a region with more or less homogenous cell population. Positive cells within the square were counted. We chose two sections of the same region from each mouse (n = 3) for each region's assessment. The means were obtained averaging the six numbers and statistics were performed as described (vide infra). The counting was done by an experimenter blind to genotype.

### Data analysis

GraphPad Prism 7.0 was used to perform Student t-tests and one-way ANOVA for evaluation of statistical differences between groups, levels are indicated as follows: *p<0.05, **p<0.01, ***p<0.001 Unless otherwise indicated, values are reported as mean ± SEM.

## Acknowledgements

We apologize to colleagues whose contributions may have been overlooked in citation of the relevant literature for this report. We thank Manuel Hernández for producing the modified open-field box for the predator odor test, to students Enrique C Guerra, Anil K Verma, Sean Sweat and Mario A Zetter for technical assistance and to Angel Fermín Barrio-Zhang for the drawing of the experimental design. LZ was a Fulbright visiting scholar to NIMH-IRP, NIH. LZ and RAB were on sabbatical stay hosted by LEE (NIMH), supported by PASPA-DGAPA-UNAM fellowships. LZ also thank Peter Somogyi and Rafael Lujan for hosting her academic research visits and discussions when part of this manuscript was developed. We thank Peter Somogyi and Rafael Lujan for critical reading and comments on an early version of this manuscript.

Grants: UNAM-DGAPA-PAPIIT-IN216918, IG200121 (LZ) and CONACYT-CB-238744 (LZ), CB-283279 (RB) and NIMH-IRP-1ZIAMH002386 (LEE).

# Additional information

## Funding

| Funder | Grant reference number | Author |
|---|---|---|
| Consejo Nacional de Ciencia y Tecnología | CB238744 | Limei Zhang |
| National Institute of Mental Health | NIMH-IRP-1ZIAMH002386 | Lee E Eiden |
| Universidad Nacional Autónoma de México | IN216918 | Limei Zhang |
| Universidad Nacional Autónoma de México | G1200121 | Limei Zhang Rafael A Barrio |
| Consejo Nacional de Ciencia y Tecnología | CB283279 | Rafael A Barrio |

The funders had no role in study design, data collection and interpretation, or the decision to submit the work for publication.

## Author contributions

Limei Zhang, Conceptualization, Resources, Data curation, Formal analysis, Supervision, Funding acquisition, Validation, Investigation, Visualization, Methodology, Writing - original draft, Project administration, Writing - review and editing; Vito S Hernandez, Conceptualization, Data curation, Software, Formal analysis, Validation, Investigation, Visualization, Methodology, Writing - review and editing; Charles R Gerfen, Resources, Data curation, Software, Writing - review and editing; Sunny Z Jiang, Data curation, Investigation, Visualization, Writing - review and editing; Lilian Zavala, Software, Formal analysis, Visualization, Writing - review and editing; Rafael A Barrio, Formal analysis, Validation, Writing - review and editing; Lee E Eiden, Conceptualization, Resources, Data curation, Formal analysis, Supervision, Funding acquisition, Validation, Investigation, Methodology, Project administration, Writing - review and editing

## Author ORCIDs

Limei Zhang ⬤ https://orcid.org/0000-0002-7422-5136
Vito S Hernandez ⬤ https://orcid.org/0000-0002-1486-1659
Rafael A Barrio ⬤ http://orcid.org/0000-0003-0987-0785
Lee E Eiden ⬤ https://orcid.org/0000-0001-7524-944X

## Ethics

Animal experimentation: This study was performed in strict accordance with the recommendations in the Guide for the Care and Use of Laboratory Animals of the National Institutes of Health. All experiments were approved by the NIMH Institutional Animal Care and Use Committee (ACUC, LCMR-08) and conducted in accordance with the NIH guidelines.

## Decision letter and Author response

Decision letter https://doi.org/10.7554/eLife.61718.sa1
Author response https://doi.org/10.7554/eLife.61718.sa2

# Additional files

## Supplementary files

- Source code 1. Matlab script for locomotion analysis.
- Transparent reporting form

## Data availability

All data generated or analysed during this study are included in the manuscript and supporting files. Source data files have been provided for Figures.

The following datasets were generated:

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

# Appendix 1

## Abbreviations

| Abbreviation | Structure name | Num. Fig.3 |
|---|---|---|
| AAA | anterior amygdalar area | 130 |
| ACAd1 | anterior cingulate area, dorsal layer 1 | 32 |
| ACAd6 | anterior cingulate area, dorsal part, layer 6 | 32 b |
| ACB | nucleus accumbens | 40 |
| AD | anterodorsal nucleus of the thalamus | 76 |
| AI | agranular insular area | 119 |
| AM | anteromedial nucleus | 55 |
| AMB | nucleus ambiguus | 146 |
| AN | cerebellar cortex, ansiform lobule, granule cell layer | 152 |
| AOB | accessory olfactory bulb | 83 |
| AON | anterior olfactory nucleus | 82 |
| AP | area postrema | 18 |
| AV | anteroventral nucleus of the thalamus | 75 |
| BAC | bed nucleus of anterior commissure | 47 |
| BLA | basolateral amygdalar nucleus | 143 |
| BMA | basomedial amygdalar nucleus | 133 |
| BST | bed nucleus of stria terminalis | 88 |
| BSTov | bed nucleus of stria terminalis, oval nucleus | 46 |
| CA2v | ventral hippocampus, CA2 | 134 |
| CA3v | hippocampal formation, ventral CA3c | 91 |
| CA3vv | Ventral tip of ventral CA3 field | 92 |
| CBgcl | granular layer of cerebellum | 90 |
| CBpj | Purkinje cell layer of cerebellum | 89 |
| CEAc | central amygdalar nucleus, capsular part | 131 |
| CEAl | central amygdalar nucleus, lateral part | 145 |
| CEAm | central amygdalar nucleus, medial part | 110 |
| CLA | claustrum | 118 |
| CLI | central linear nucleus raphe | 29 |
| COAa | cortical amygdala anterior part | 107 a |
| COAp | cortical amygdala posterior part | 107 b |
| CP | caudoputamen | 80 |
| CU | cuneate nucleus | 61 |
| DCO | dorsal cochlear nucleus | 126 |
| DG-sg | dentate gyrus, granule cell layer | 87 |
| DH | dorsal hippocampus proper | 123 |
| DMH | dorsomedial nucleus of the hypothalamus | 52 a |
| DP | dorsal peduncular área (TTd) | 34 a |
| DR | dorsal nucleus raphe | 15 |
| ECT | ectorhinal area | 138 |
| ENTl | entorhinal area, lateral part | 139 |
| ENTm | entorhinal area, medial part | 140 |

| Abbreviation | Structure name | Num. Fig.3 |
|---|---|---|
| EP | endopiriform nucleus | 117 |
| EW | Edinger-Westphal nucleus | 12 |
| FL | cerebellar cortex, flocculus, granule cell layer | 151 |
| FN | fastigial nucleus of the cerebellum | 60 |
| FS | fundus of striatum | 116 |
| GPe | globus pallidus, external segmet | 114 |
| GPi | globus pallidus, internal segment (entopeduncular nucleus) | 115 |
| GRN | gigantocellular reticular nucleus | 24 |
| GU | gustatory areas | 136 |
| IA | intercalated amygdalar nucleus | 132 |
| IC | inferior colliculus | 17 |
| IF | nucleus raphe, interfascicular | 11 |
| IGL | intergeniculate leaflet of the lateral geniculate complex | 150 |
| ILA | infralimbic area | 42 |
| IO | inferior olivary complex | 21 |
| IPN | interpeduncular nucleus | 28 |
| IRN | intermediate reticular nucleus | 64 |
| LA | lateral amygdalar nucleus | 144 |
| LC | locus coeruleus | 65 |
| LGd | dorsal part of the lateral geniculate complex | 149 |
| LGv | ventral part of the lateral geniculate complex | 135 |
| LM | lateral mammillary nucleus | 71 |
| LPO | lateral preoptic area | 73 |
| LSc | lateral septal nucleus caudal | 39 |
| LSr | lateral septal nucleus rostral | 36 |
| MARN | magnocellular reticular nucleus | 67 |
| MBO | mammillary body | 8 |
| MD | mediodorsal nucleus of the thalamus | 6 |
| MDRN | medullary reticular nucleus | 68 |
| MEAad | medial amydgala, anterodorsal part | 111 |
| MEApd | medial amygdala, posterodorsal part | 112 |
| MEApv | medial amygdala, posteroventral part | 105 |
| MEPO | median preoptic nucleus | 3 |
| MG | medial geniculate complex | 99 |
| MH | medial habenula | 45 |
| MOB | main olfactory bulb | 1 |
| MOBgl | main olfactory bulb, glomerular layer | 85 |
| MOBgr | main olfactory bulb, granule layer | 84 |
| MOBml | main olfactory bulb, mitral layer | 86 |
| MOs | secondary motor area | 44 |
| MoV | motor nucleus of trigeminal nerve | 94 |

| Abbreviation | Structure name | Num. Fig.3 |
|---|---|---|

*Continued on next page*

*continued*

| Abbreviation | Structure name | Num. Fig.3 |
|---|---|---|
| MPO | medial preoptic area | 48 |
| MRN | midbrain reticular nucleus | 69 |
| MS | medial septal nucleus | 37 |
| MV | medial vestibular nucleus | 59 |
| NDB | diagonal band nucleus | 38 |
| NLL | nucleus of lateral lemniscus | 97 |
| NLOT | nucleus of the lateral olfactory tract | 108 |
| NTS | nucleus of the tractus solitarius | 19 |
| OP | olivary pretectal nucleus | 147 |
| ORB | orbital area | 31 |
| OT | olfactory tubercle | 41 |
| OV | vascular organ of the lamina terminalis | 4 |
| PA | posterior amygdalar nucleus | 106 |
| PAA | piriform-amygdalar area | 141 |
| PAG | periaqueductal gray | 14 |
| PAR | parasubiculum | 128 |
| PB | parabrachial nucleus | 63 |
| PBG | parabigeminal nucleus | 98 |
| PCG | pontine central grey | 58 |
| PF | parafascicular nucleus | 79 |
| PG | pontine grey | 27 |
| PH | posterior hypothalamic nucleus | 10 |
| PHA | posterior hypothalamic area | 74 |
| PIR | piriform area | 129 |
| PL | prelimbic area | 43 |
| PM | premammillary nucleus | 148 |
| POST | postsubiculum | 124 |
| PP | peripeduncular nucleus | 101 |
| PRC | precommisural nucleus | 78 |
| PRE | presubiculum | 125 |
| PRNr | pontine reticular nucleus | 25 |
| PSV | principal sensory nucleus of trigeminal nerve | 95 |
| PT | parataenial nucleus | 56 |
| PTLp | posterior parietal association areas | 121 |
| PVH | paraventricular hypothalamic nucleus | 49 |
| PVp | periventricular hypothalamic nucleus, posterior part | 53 |

| Abbreviation | Structure name | Num. Fig.3 |
|---|---|---|
| PVT | paraventricular nucleus of the thalamus | 5 |
| RE | nucleus of reuniens | 7 |
| RL | rostral linear nucleus raphe | 13 |
| RM | nucleus raphe magnus | 23 |

*Continued on next page*

*continued*

| Abbreviation | Structure name | Num. Fig.3 |
|---|---|---|
| RPA | nucleus raphe pallidus | 22 |
| RSP | retrosplenial area | 33 |
| RT | reticular nucleus of the thalamus | 77 |
| rV | trigeminal reticular nucleus | 26 |
| SCH | suprachiasmatic nucleus | 50 |
| SCm | superior colliculus, motor related | 16 |
| SCs | superior colliculus, sensory related | 57 |
| SFO | subfornical organ | 2 |
| SI | substantia innominata | 81, 113 |
| SN | substantia nigra | 70 |
| SO | supraoptic nucleus | 51 |
| SOC | superior olivary complex | 66 |
| SPF | subparafascicular nucleus | 100 |
| SPIV | spinal vestibular nucleus | 62 |
| SpV | spinal nucleus of trigeminal | 96 |
| SS | somatosensory areas | 120 |
| STN | subthalamic nucleus | 103 |
| SUB | subiculum | 104 |
| SUM | supramammillary nucleus | 9 |
| TR | postpiriform transition area | 142 |
| TT | taenia tecta | 34b |
| TTv | taenia tecta ventral | 35 |
| UVU | uvula of cerebellum | 30 |
| VCO | ventral cochlear nucleus | 127 |
| vHilus | ventral hilus | 93 |
| VIS | visual areas | 122 |
| VISC | visceral area | 137 |
| VLH | ventrolateral hypothalamic nucleus | 72 |
| VMH | ventromedial hypothalamic nucleus | 52 b |
| VTA | ventral tegmental area | 54 |
| XII | hypoglossal nucleus | 20 |
| ZI | zona incerta | 102 |

Color indicate embryological origin according to Allen Brain Atlas Common Coordinate Framework as follows:

| cortical plate | subcortical plate | cerebral nuclei | interbrain |
|---|---|---|---|
| midbrain | hindbrain | cerebellum | |

