## [Decision Letter]

**Acceptance summary:**

The reviewers find this manuscript to be of significant value to the field, as it provides a comprehensive and nicely presented map of PACAP and PAC1 expression across the brain, including novel and detailed information on the cell types in which expression was found.

**Decision letter after peer review:**

Thank you for submitting your article "Behavioral role of PACAP reflects its selective distribution in glutamatergic and GABAergic neuronal subpopulations" for consideration by *eLife*. Your article has been reviewed by two peer reviewers, and the evaluation has been overseen by a Reviewing Editor and Ronald Calabrese as the Senior Editor. The following individuals involved in review of your submission have agreed to reveal their identity: Tim James Viney (Reviewer #1); Gilliard Lach (Reviewer #2).

The reviewers have discussed the reviews with one another and the Reviewing Editor has drafted this decision to help you prepare a revised submission.

We would like to draw your attention to changes in our revision policy that we have made in response to COVID-19 (https://elifesciences.org/articles/57162). Specifically, when editors judge that a submitted work as a whole belongs in *eLife* but that some conclusions require a modest amount of additional new data, we are asking that the manuscript be revised to either limit claims to those supported by data in hand, or to explicitly state that the relevant conclusions require additional supporting data.

Summary:

The manuscript of Zhang and colleagues studied the expression of PACAP and PAC1 mRNA in inhibitory and excitatory neurons in the entire mouse brain by using dual ISH method. Additionally, a behavioural test is carried out to provide a functional role for PACAP/PAC1 on olfaction and defensive behaviour followed by cFos examination of selected brain regions to indicate the role of PACAP and PAC1 in such behavioural outputs. The reviewers believe that this is a valuable and important study on PACAPergic brain regions in mice, especially relating to the hypothalamus, but would benefit from a major reorganisation to improve the presentation of data, and further quantitative criteria to strengthen the observations. Please address the considerable comments detailed in this letter in your revised submission.

Reviewer #1:

1) To appeal to the more general readership of *eLife*, the paper would benefit from a reorganisation, especially when referring to figures and tables. There are a very large number of abbreviations. A list near the beginning of the manuscript would help the reader, and would also shorten the figure legends and improve readability/flow. For the non-expert, some areas should be labelled/highlighted separately or provide more information in the figures, e.g. “ACA and the entorhinal cortex” one has to search the figure legend, find the number then search the figure panels to find the location of these brain regions. Abbreviations and brain region names should be consistent, e.g. ACC is used in text, but ACA in figure and legend. Unless mistaken, Table S1 is not mentioned in the text. Figure 9 is first mentioned in the Discussion. Since these are valuable data, refer to this figure in the main Results section in terms of the knockout. Figure 1—figure supplement 1 is very informative, but requires a lot of searching to find the panel that is referred to in the text. In Figure S1-7/7-M, panels M1-4 are identical to Figure 1E-H and the scale bar in M3 is different to 1G.

2) In several places there are anecdotal statements and it is not clear about the reproducibility of the results. The methods for quantification (including those mentioned in table legends) should be included in Materials and methods.

For animals, please check and state the total number of mice and rats used in the study, and whether EGFP mice were also used.

For c-fos experiments, how were these cells counted, how many sections per mouse, what was the section thickness, how were the values calculated (mean, absolute numbers). Was fos counting done blind to genotype?

Was there variation between animals in terms of expression levels/strength? Case/animal numbers in figures would help. It is not clear what is meant throughout by statements such as “strongest”. Is this by density in cells or number/intensity of puncta? For example, section 3.1, retina. What is meant by “higher percentage than previously reported”? Is this referring to both previous reports in mice? Also see Engelund et al. Cell Tissue Res 2010. How many samples and/or mice were examined and how were ganglion cells counted?

Similarly, cortical expression in different layers, how were the values of 80% obtained? Again, “highest expression level of PAC1 among all brain regions” is a strong claim, how was this quantified? Subsection “The claustrum (CLA)”, need references/evidence for observations of mouse claustrum percentages. “more than 90%”. “the highest expression of PACAP was observed in the MnPO”.

In terms of the olfactory pathways, is there evidence of co-transmission or is this a hypothesis?

Some claims will need careful revision. E.g. in the Figure 5 legend, the last sentence contradicts the main text.

The finding that 100% of the 3 GABAergic subpopulations expressed PAC1 is a big claim, yet there is no quantification to back this up. How many brain regions were examined, how many mice, sections, counted cells etc.? If it just refers to primary somatosensory cortex, was it all or some layers?

Table 2 (also applies to parts of Table 1), do blank areas of the table mean not examined? Or should there be “-“ in these areas? For example, the medial septal complex contains vglut2 expressing cells but the corresponding row/column is blank.

There is the claim that PACAP mRNA was not found in cell body layers, but in Table 1 it is reported that there is weak expression in VGLUT1^+^ cells. Since VGLUT1 cells are in the pyramidal cell layer, this seems contradictory. It would be helpful to have a higher power image of CA1 (as for rat in Figure S2). Could expression outside this layer be in subpopulations of GABAergic neurons? Were these examined (blank in Table 1)? DG is also missing from Table 1.

PAC1 expression. Subsection “Hippocampal formation” claims it is selective for VGAT cells. But there are clear examples of VGAT- cells in Figure S3B expressing PAC1. What are these?

3) Suggestion about paracrine/autocrine signalling. Is there is evidence in literature for such a role? This seems speculative without immunohistochemical evidence. Hannibal 2002, carried out at both the protein and mRNA levels, showed axon terminals in multiple regions. Can these be mapped to the regions that express PAC1 in mice? Is there any evidence or could the authors comment on the existence of presynaptic PACAP receptors? Expression of PAC1 mRNA does not imply that the cell would express the protein exclusively along its somatodendritic membrane. “Classical” neurotransmission presumably could occur in PACAP/PAC1 rich regions via local axons in addition to long-range axons.

4) The observation of PACAP in part of temporal CA3, which the authors refer to as CA3c, has in fact previously been defined as CA3vv, corresponding to the coch expressing domain (see Thompson et al., 2008, Fanselow and Dong, 2010). PACAP may indeed be an additional marker along with calretinin for this principal cell subpopulation, and they may want to revise their model or refer to these earlier papers.

5) PACAP KO. Some clarification would be welcome in terms of animal cohorts. Please state the experimental unit (i.e. n=9 mice/group). In D, the freezing data show only 8 mice, was one pair excluded due to lack of freezing in an animal, as for jumping mice in C? In Ai, Aii, Bi, Bii, does this show the traces for the total time?

In the separate experiment, was n=3 a separate cohort of mice or from the N=18 total as stated in the Materials and methods? Is the n=3 per group or total mice? This may require an increase sample size for this claim, or show quantification/statistical test. For this test, were experimenters also blind to the genotype? The last sentence is difficult to follow.

For the behavioural tests, please include details about whether the wooden boxes, room and experimenter were familiar to the mice before the test (which could affect variability), whether mice were tested at the same time of day, and if KO and WT animals were housed together.

In the Discussion, can the authors comment on or provide evidence of possible developmental changes / compensatory mechanisms occurring in the KO animals.

Reviewer #2:

Part 1: the PACAP/PAC1 characterization is well designed and executed. The result description is lengthy and sometimes confusing. Figures and tables (including the supplementary information) are clear and informative. The authors decide to do not show Vipr1/Vipr2 data, which should be reconsidered for publication. Overall, this part of the manuscript represents a nice piece of work and surely will be very helpful to whom wish to work with PACAP/PAC1.

Part 2: I think this part is the critical one in this manuscript. Starting from section 4, it uses the part 1 of the manuscript to review the literature and build a neuronal circuit with PACAP/PAC1 that makes for behavioural processes. It is literally a review inside the Results section. The schematic figures are interesting but also quite speculative regarding brain signalling since the authors did not performed any experiment to investigate the pathway of PACAP and the literature is scarce. Moreover, the role of Vip receptors were completely neglected here.

---

## [Author Response]

Revisions for this paper:Reviewer #1:1) To appeal to the more general readership of eLife, the paper would benefit from a reorganisation, especially when referring to figures and tables. There are a very large number of abbreviations. A list near the beginning of the manuscript would help the reader, and would also shorten the figure legends and improve readability/flow. For the non-expert, some areas should be labelled/highlighted separately or provide more information in the figures, e.g. “ACA and the entorhinal cortex” one has to search the figure legend, find the number then search the figure panels to find the location of these brain regions. Abbreviations and brain region names should be consistent, e.g. ACC is used in text, but ACA in figure and legend.

We thank the reviewer for the appreciation and careful and professional criticism to benefit clarity in presenting our results to the readership of *eLife*.

The inconsistencies of term usage (produced by using Allen brain maps and the Paxino's and Swanson atlas) have been corrected and now we unified the nomenclature usage to *Allen Brain maps (http://mouse.brain-map.org/).* The related difficulties on the aspect of *abbreviations vs full anatomical region's names* mentioned by Reviewer #1 have been emended by inserting a table of nomenclatures number and *embryological origin* color-coded, according to the *Allen Brain Atlas Common Coordinate Framework*. The table is now ordered in alphabetical order, with full region's abbreviations being the first column from the left, which are the ones used in the entire text and the figure legends, in order to facilitate the search while saving space. Besides, we made the number codes column for Figure 3.

Unless mistaken, Table S1 is not mentioned in the text.

Actually, it was mentioned in the first submitted version. We have made some corrections on this table: a) in the title we added the word "selective", i.e. "Density distribution of PAC1 expressing cells in *selective* cortical regions; b) we unified the abbreviations of the brain regions studied to be consistent with the new nomenclature table in the main text (however listed in the table caption since it is an independent file); c) we explain the reason why those regions were chosen for our sampling in the table legends; d) the table is now called Figure 3—source data 3, due to the addition of two more tables for Vipr1 and Vipr2.

Figure 9 is first mentioned in the Discussion. Since these are valuable data, refer to this figure in the main Results section in terms of the knockout.

We are embarrassed for this omission that occurred during versions passing among authors and changing from word (.docx) version to Overleaf software for LaTex for *eLife* template, where an old version was already uploaded there, that in some moment this paragraph was missed unwilling and also the text was moved upwards, above its corresponding subtitle "2" (now returned to its corresponding place). We sincerely apologize to the reviewers for not realizing this before the first submission, which might have cause confusions. These are all emended. The paragraph referring to Figure 9 is added at the very end of the Results section:

"We assessed *fos* expression in the PACAP/PAC1 system including both glutamatergic and GABAergic co-expressing neurons in response to odorant exposure, using DISH technique and VGLUT1, VGLUT2 and VGAT mRNAs probes to identify the glutamatergic and GABAergic neurons. Surprisingly, we observed a sharp reduction in the abundance of three vesicular transporters in the main PACAP containing nuclei we described (Figure 9). This reduction was observed both as reduced abundance of the ISH staining puncta at the single cell level and the density of expressing cells in the given region (Figure 9 E)."

Figure 1—figure supplement 1 is very informative, but requires a lot of searching to find the panel that is referred to in the text. In Figure S1-7/7-M, panels M1-4 are identical to Figure 1E-H and the scale bar in M3 is different to 1G.

We thank the reviewer once more for this careful examination. We have revised the entire text aiming to make the citations more precise. We have added the number ID of regions from the figure 3, to the text when they were mentioned. We would also like to explain that we kept the duplicated panels thinking that the Figure 1—figure supplement 1, which is a comprehensive mapping by itself, could be used independently for readers for anatomical reference. The Figure 1E-H were taken from this figure as examples of the DISH technique demonstration. In other words, they are the same pictures but aiming to show different aspects in two different places; hence, for SI-Figure 1's comprehensiveness we would like to keep them in both files. We have added a sentence in the figure legend of Figure 1 “Note: this figure contains excerpts from the more comprehensive Figure 1—figure supplement 1”.

We thank the reviewer for pointing out the inconsistence of the scale bars – the M3 was wrongly labelled as "100µm" and is now corrected to "250µm". We have also made a thorough revision/edition of this figure, all the panels, mainly adding the missing region labels and correcting the wrongly located arrows and completed the figure captions with all the abbreviations defined in accordance with the abbreviation table of the main text.

2) In several places there are anecdotal statements and it is not clear about the reproducibility of the results. The methods for quantification (including those mentioned in Table legends) should be included in Materials and methods.

We have corrected anecdotal statements as requested. The deleted segments are listed as appendix of this document.

We have added the following sentences in the Materials and method section, as well as the beginning of the "Results" sections and have unified the descriptions in the whole text using the terms "weak", "low", "moderate", "intense" and "very intense" (labelled in italics) only, to give a semiquantitative notion for each given description.

"For semi-quantitative scoring we used similar criteria from literature (Hannibal, 2002). Concretely, we used the annotation criteria as the following: the percentage of expressing cell/total Nissl stained nuclei: “-”, not observed; “+”, weak (<20%); “++”, low (20%-40%); “+++”, moderate (40%-60%); “++++”, intense (60%-80%); “+++++”, very intense (>80%)".

For animals, please check and state the total number of mice and rats used in the study, and whether EGFP mice were also used.

We have now added clearly in Materials and methods the details for quantification for both figures and tables and numbers of animals used. This study did not include EGFP mice (the sentence in the text was a reference to the work of Condro et al. and is now so indicated). Text yext refers to the total number of mice used for distribution studies by DISH analysis.

For c-fos experiments, how were these cells counted, how many sections per mouse, what was the section thickness, how were the values calculated (mean, absolute numbers). Was fos counting done blind to genotype?

We thank the reviewer for raising this question which we have missed to specify in the previous version. The following paragraph is now added to the text:

"The counting for fos and transporters expressed cells was done on a computer screen connected to a digital camera mounted over a light microscope. The region of interest (ROI) was centred through observation using the microscope oculars and 20x objective and projected to a large computer screen through a digital camera. A fixed square equivalent to 0.0314mm^2^ for the magnification/computer enlargement was pre-fixed and moved to the ROI choosing a region with more or less homogenous cell population. Positive cells within the square were counted. We chose 2 sections of the same region from each mouse (n=3) for each region's assessment. The means were obtained averaging the 6 numbers and statistics were performed as described (vide infra). The counting was done by an experimenter blind to genotype *(see acknowledgement)."*

The thickness of the section was 12 µm as specified in the text.

Was there variation between animals in terms of expression levels/strength? Case/animal numbers in figures would help. It is not clear what is meant throughout by statements such as “strongest”. Is this by density in cells or number/intensity of puncta? For example, section 3.1, retina. What is meant by “higher percentage than previously reported”? Is this referring to both previous reports in mice? Also see Engelund et al. Cell Tissue Res 2010. How many samples and/or mice were examined and how were ganglion cells counted?

These definitions such as "strongest", "moderate" were always referring the number of cells against the total of Nissls stained nuclei as we first mentioned in the table legend of Table 1 in the first version and not referred the number of puncta. We have implemented this in the text added to the Results and Materials and methods sections and we avoided the usage of "strong" or "strongest". Instead, we are now using "intense" and "very intense" to be in accordance with the quantification criteria we stated.

See above for your concern on case/animal numbers.

The retina data was reported in a previous publication from Eiden's group and collaborators that we cited in this study only for the sake of completeness. We cited the full reference.

Similarly, cortical expression in different layers, how were the values of 80% obtained? Again, “highest expression level of PAC1 among all brain regions” is a strong claim, how was this quantified? Subsection “The claustrum (CLA)”, need references/evidence for observations of mouse claustrum percentages. “more than 90%”. “the highest expression of PACAP was observed in the MnPO”.

See the reply above: we have unified the description to only use the terms *"weak", "low", "moderate", "intense" and "very intense" (labelled in italics),* which correspond to semiquantitative percentage stated in the beginning of the Results and in the Materials and methods sections.

In terms of the olfactory pathways, is there evidence of co-transmission or is this a hypothesis?

We thank the reviewer again for pointing out this particular anecdotal-style mention. Here, we referred to co-transmission of PACAP/PAC1 with VGAT or VGLUT1/VGLUT2/VGLUT3 that we have described in the previous section. However, the glutamate/GABA co-transmission, a point not intrinsically related to this report is also reported in Table 1. For instance, the neurons in the outer plexiform layer of the main olfactory bulb (MOB), besides expressing PACAP mRNA, expressed mRNAs of VGLUT1(+++), VGLUT2 (+++) and VGAT (+). We cannot assure that all the mRNAs could co-localize within the same cell, as for the DISH method detection. We did test that some of cells co-expressed a given VGLUT and VGAT. It is interesting to note that vasopressin mRNA had similar co-expression pattern as PACAP in these cells (see Zhang et al., JNE 2020), which suggest co-expression of these two neuropeptides, together with small neurotransmitters, similar to the case in the magnocellular neurons of the hypothalamic paraventricular nucleus extensively reported.

We modified the first sentence of the paragraph referred by adding: "The olfactory system appears specially to use PACAP/PAC1 as one of its main modes of co-transmission (see section 3. 2. 1), especially within the cell population in the outer plexiform layer of the MOB and in the mitral cell layer of the AOB, in which some cells were observed to co-express VGLUT1/VGLUT2 and VGAT". We deleted the long sentence which did not belong to this result section.

Some claims will need careful revision. E.g. in the Figure 5 legend, the last sentence contradicts the main text.

We thank the reviewer for noting this discrepancy/typographical error. In the Figure 5 legend we meant to say "… Slc17a7 (F), and *(not "but")* we did not (*not "also")* observe co-expression within the Slc32a1 cells (G)". It is now corrected. The panel was aimed to show "no-co-expression" which was inserted under that panel.

The finding that 100% of the 3 GABAergic subpopulations expressed PAC1 is a big claim, yet there is no quantification to back this up. How many brain regions were examined, how many mice, sections, counted cells etc.? If it just refers to primary somatosensory cortex, was it all or some layers?

We thank the reviewer for pointing out this paragraph that lacks specifications. We have amended the paragraph. The added words to limit our conclusion are labelled in bold:

"PAC1 mRNA expression in neocortex was widespread with more homogenous aspects concerning the different cortices, except that in the ACA and the entorhinal cortex layers 2-3 and layers 5-6, which showed *very intense* expression levels (Figure 3, panels B and F) and https://gerfenc.biolucida.net/images?selectionType=collection&selectionId=98. We sampled eight neocortex regions, at two coronal levels, Bregma 0.14mm and Bregma 1.7mm, where we observed that more than 80% of neurons in layers 2-3 and layer 5 expressing PAC1 mRNA (Figure 3—source data 3). As approximately 20% of cortical neurons were GABAergic (Petilla Interneuron Nomenclature, Ascoli et al., 2008), we tested the three main GABAergic cell types in these cortical regions, finding that all of somatostatin (Sst), parvalbumin (PV)and corticotropin releasing hormone (CRH) neurons in the regions we sampled co-expressed PAC1 (Figure 3—figure supplement 3F, G, H)".

We replaced the expression "100%" with "all", which we meant the VGAT neurons, in the fields we examined, *all* co-expressed PAC1 (see the SI-Tab. 3 caption). In most of the cases there were less than 10 VGAT neurons within the ROI of (0.03mm2), so using a percentage for such small numbers was just not adequate. We are sorry for not having considered this aspect that made confusions.

Table 2 (also applies to parts of Table 1), do blank areas of the table mean not examined? Or should there be “-“ in these areas? For example, the medial septal complex contains vglut2 expressing cells but the corresponding row/column is blank.

Corrected this aspect in all the tables and made definitions for abbreviations. We thank the reviewer for pointing this out.

There is the claim that PACAP mRNA was not found in cell body layers, but in Table 1 it is reported that there is weak expression in VGLUT1^+^ cells. Since VGLUT1 cells are in the pyramidal cell layer, this seems contradictory. It would be helpful to have a higher power image of CA1 (as for rat in Figure S2). Could expression outside this layer be in subpopulations of GABAergic neurons? Were these examined (blank in Table 1)? DG is also missing from Table 1.

We thank the reviewer again for this careful examination and we apologise again for not revising carefully the table contents. Specifically, those 4 "+" referred to by the reviewer, corresponded to the column of Hannibal, 2002. This has been corrected. The shared-first authors of this work, LZ and VSH, returned to revise each region and their semiquantitative scores and we found, in dorsal hippocampus, there are clear PACAP mRNA expression in the CA2 region. Since this is an important discovery, we have scored "+++" in CA2 and inserted photomicrographs in the Figure 5C as insets. This sentence was added to the text: "However, we report here the marked and selective expression of PACAP in pyramidal neurons of the CA2 region (Figure 4C insets) and in the hilus (vide infra)".

We have also made the following modifications to Table 1: (1) the MOB two of the strong PACAP expressing cell groups were wrongly identified as periglomerular cells – they are corrected as outer plexiform cells and periglomerular cells with different expressing densities; (2) five missing regions were addressed, i. e. dorsal DG and ventral DG, parasubthalamic nucleus and posterior hypothalamic nucleus, and pontine central grey; (3) some expression strengths were corrected (highlighted in the labelled version); (4) we have simplified Table 1 by removing the column for VGLUT3 expression of which there were only a few instances.

PAC1 expression. Subsection “Hippocampal formation” claims it is selective for VGAT cells. But there are clear examples of VGAT- cells in Figure S3B expressing PAC1. What are these?

The term “selective expression” of PAC1 mRNA in VGAT-positive cells in CA subfields in the first version is meant to convey that most PAC1-expressing cells in CA subfields are VGAT-positive, while the (far more numerous; presumptively excitatory) neurons are *mainly* non-PAC1-positive. It is evident that there were two cells in the referred field that expressed PAC1 but clearly not VGAT that we have not identified. By the way, this figure and panel is now Figure 3—figure supplement 4 because of the addition of Vipr1 and Vipr2 that have been called S3 and S4.

3) Suggestion about paracrine/autocrine signalling. Is there is evidence in literature for such a role? This seems speculative without immunohistochemical evidence. Hannibal, 2002, carried out at both the protein and mRNA levels, showed axon terminals in multiple regions. Can these be mapped to the regions that express PAC1 in mice? Is there any evidence or could the authors comment on the existence of presynaptic PACAP receptors? Expression of PAC1 mRNA does not imply that the cell would express the protein exclusively along its somatodendritic membrane. “Classical” neurotransmission presumably could occur in PACAP/PAC1 rich regions via local axons in addition to long-range axons.

According to modern neuroendocrinology the autocrine, paracrine mechanisms are defined by the co-expression of a given neuropeptide and its receptor(s) or expression of its receptor in the adjacent cells, since the soma-dendritic release of neuropeptides are widely documented (for recent reviews see Leng, 2018; Colin H Brown, Mike Ludwig, Jeffrey G Tasker, Javier E Stern. Somato-dendritic vasopressin and oxytocin secretion in endocrine and autonomic regulation. J. *Neuroendocrinol*. 2020 Jun;32(6):e12856. doi: 10.1111/jne.12856. Epub 2020 May 14.). In our case, we suggested the autocrine/paracrine mechanisms for PACAP/PAC1 signalling only at DISH level, by showing that PACAP and PAC1 mRNAs were expressed withing the same cells and their adjacent cells, as we showed in Figure 3. Please note that this notion does not exclude that the same peptide and its receptor(s) can use the neurocrine mechanisms (secretion from synaptic cleft), even within the same cell. We just wanted to emphasise that we are aware of all those possibilities, but we consider it is merited the demonstration, at DISH level, the notion for autocrine/paracrine mechanisms for PACAP/PAC1 signalling.

4) The observation of PACAP in part of temporal CA3, which the authors refer to as CA3c, has in fact previously been defined as CA3vv, corresponding to the coch expressing domain (see Thompson et al., 2008, Fanselow and Dong, 2010). PACAP may indeed be an additional marker along with calretinin for this principal cell subpopulation, and they may want to revise their model or refer to these earlier papers.

We now reference the detailed genomic “map” of mouse hippocampus of Thompson et al., 2008, as reviewed by Fanselow and Dong, 2010, to put in context our observation of PACAP-positive neurons of CA3c as likely CA3vv, although we have not co-stained with Coch mRNA to confirm this. We thank the reviewer for helping us to give this point the attention it deserves.

5) PACAP KO. Some clarification would be welcome in terms of animal cohorts. Please state the experimental unit (i.e. n=9 mice/group). In D, the freezing data show only 8 mice, was one pair excluded due to lack of freezing in an animal, as for jumping mice in C? In Ai, Aii, Bi, Bii, does this show the traces for the total time?

The subjects for behavioural assessment (n=9) were all from the same cohort (performed in March 2019, during a study visit that the two first authors lead the experiments, performed in NIMH, with duly permissions from our institutions). The experiments for the double odour stimulus (n=3, performed in June 2019 in NIMH, led by LZ and VH) and for the olfaction cookie retrieval test (n=3 for each gender, N=12, performed oct 2020) were from other cohorts. Besides, due to the incomplete documentation of the experiment of June 2019 (LZ and VH visiting NIMH) and the current difficulty to repeat this ancillary experiment, we have deleted the mention of former experiment (n=3, double amount of odour stimulus produced higher freezing behaviour in both groups) and replaced with the new experiment of cookie retrieval.

The reason for the n=8 in behavioural assessment analysis for freezing is because that two outliers, one in each experimental group, were excluded. We have added the clarification at the end of the figure legend of Figure 8: "Note that during the freezing analysis, two outliers, one in each experimental group were discarded because being 45 and 24 (counts), with 5 times and 11 times higher than the standard deviations for the corresponding groups, i.e. WT:14.29 ± 6.17 (mean ± SD) vs KO: 3.18 ± 1.80 (mean ± SD). This exclusion was based on criteria explained in NIH Rigor and Reproducibility Training course, https://www.nigms.nih.gov/training/documents/module4-sample-size-outliers-exclusion-criteria.pdf and http://www.itl.nist.gov/div898/handbook/prc/section1/prc16.htm."

In the separate experiment, was n=3 a separate cohort of mice or from the N=18 total as stated in the Materials and methods? Is the n=3 per group or total mice? This may require an increase sample size for this claim, or show quantification/statistical test. For this test, were experimenters also blind to the genotype? The last sentence is difficult to follow.

In light of the importance to establish that PACAP-deficient mice are capable of detecting odors, we performed a more explicit simple behavioral assessment of olfaction in these mice (Yang and Crawley, Current Protocols in Neuroscience 8.24.1-8.24-12, July 2009). We have added the following description in place of our previous description of olfaction-testing in wild-type versus PACAPko mice: “An equal number of male and female mice (n=6 wild-type mice, 3 each male and female, and n=6 PACAPko mice, 3 each male and female) were tested for the ability to retrieve a buried cookie according to the procedure described by Yang and Crawley, Current Protocols in Neuroscience 8.24.1-8.24-12, July 2009. Briefly, mice were individually housed under standard light-dark conditions for a period of four days. Subsequently, a cookie (Chocolate Teddy Grahams—Nabisco) was placed in each cage at 11:45 a.m. (i.e. halfway through the day of the standard 12:12 light/dark cycle). The following day, cookie consumption was confirmed, and a second cookie placed in each cage, consumption confirmed by end of day, and all food removed at approximately 4 p.m. The next day, mice were transported to the testing room, with cages re-labeled for the blinded observer at 10:45 a.m. One hour later, testing was initiated by placing mice in a fresh cage (test cage) with bedding 3 cm deep. Mice were allowed to explore the cage for 5 min; removed to a fresh cage while a cookie was buried 1 cm deep at one end of the test cage, and the mouse placed at the opposite end to the buried cookie in the test cage. Time for mouse to retrieve the buried cookie and hold the cookie in its forepaws and begin to eat was recorded. Mice were tested in pairs in a biosafety hood containing two pairs of test/holding cages at a time; testing was completed by 1:45 p.m. (3 hours after initial transport of mice to testing room). Retrieval times for wild-type and PACAPko mice were not statistically significantly different (wild-type retrieval latency 38.0 +/- 10.4 seconds; PACAPko retrieval latency 44.5 +/- 12 seconds, means +/- s.e.m., n=6/group).

We believe that the use of a separate cohort of mice, using a standard test for olfaction, clarifies the issue of whether or not anosmia or hypo-osmia might complicate the interpretation of the experiments described for response to predator odor: we conclude that it does not.

For the behavioural tests, please include details about whether the wooden boxes, room and experimenter were familiar to the mice before the test (which could affect variability), whether mice were tested at the same time of day, and if KO and WT animals were housed together.

All behavioural experiments were performed in the Institutes animal facility's experiment room, next to their usual residential room. The mice were housed 4-5 per cage. The wooden box was located within a functioning hood with low noise level and dim light. The experimental subjects were introduced for the first time to the environment (no previous habituation). All were tested between 10am-2pm of the same day.

In the Discussion, can the authors comment on or provide evidence of possible developmental changes / compensatory mechanisms occurring in the KO animals.

We cannot presently comment on whether developmental changes or compensatory mechanisms occur in PACAP KO mice: we have embarked upon detailed transcriptomic changes in PACAPko compared to WT mice to address this issue. To do this, we have developed several lines of conditional PACAP knock-out mice and are breeding them with various Cre-driver lines to assess (a) basal transcriptome differences and (b) whether these are affected by PACAP ablation at different developmental stages. This very interesting question will be the subject of future reports in which we will reference this initial report, and ascribe transcriptomic changes noted here either to “constitutive”/developmental effects of PACAP loss, or acute effects contingent upon neuronal activation as evidenced by fos expression. At this time, we can say that for PACAPergic neurons activated by predator odour exposure, there is down-regulation, in a regiospecific way, of either VGAT (e.g. ansiform lobule of cerebellum) or VGluT1 (e.g. PFC) in fos-activated PACAPergic neurons. We hypothesize this effect as occurring as a result of PACAP release in these areas, but cannot specify the mechanisms of regulation until we have completed the developmental studies sketched out above.

Reviewer #2:Part 1: the PACAP/PAC1 characterization is well designed and executed. The result description is lengthy and sometimes confusing. Figures and tables (including the supplementary information) are clear and informative. The authors decide to do not show Vipr1/Vipr2 data, which should be reconsidered for publication. Overall, this part of the manuscript represents a nice piece of work and surely will be very helpful to whom wish to work with PACAP/PAC1.

We first would like to thank the reviewer for the very careful and detailed revision and criticisms, which helped us to improve the quality of this manuscript. As for the first concern, please kindly see our replies to Reviewer 1 and the appendix listing the deleted segments that were not indispensable and rendered the previous version "lengthy and sometimes confusing".

For Vipr1 and Vipr2 expression mapping, upon receiving your request, we have added two tables and two figures (photo galleries) to the supplementary information to provide a more comprehensive description, that we think they can nicely complement the information presented in the main manuscript. However, we also added these two lines in the new version: " To simplify this already extensive report, we present the data for these two receptors in Figure 3—figure supplements 1 and 2 and source data 1 and 2. " and deleted the specific region description from the previous version: "Vipr1 (VPAC1) expression was widespread, like PAC1 mRNA, in cerebral cortex, hippocampal formation (prominently in the mossy cells), structures derived from cerebral subplate and cerebral nuclei, as well as hypothalamus. The cerebellar cortex in the flocculus and deep cerebellar nuclei moderately expressed, and the Purkinje cells strongly expressed Vipr1; Vipr2 (VPAC2) was observed to be strongly and selectively expressed in the MOB, the mitral and granule layers, the BNSTov, CEAc, lateral division, the SCN, ventral anterior, posterior, posterior medial and lateral geniculate nuclei of thalamus, hypothalamic preoptic area, suprachiasmatic nucleus, inferior, midbrain inferior colliculus, interpeduncular nucleus, periaqueductal gray, and superior colliculus, , hindbrain dorsal tegmental nucleus, cranial nerve nuclei III, V nucleus of the lateral lemniscus, pontine reticular nucleus, superior olivary complex, nucleus raphé pontis, and paragigantocellular reticular nucleus in medulla the nucleus of the trapezoid body and the facial motor nuclei showed a high expression, and in the cerebellum the paraflocculus and flocculus granule and molecular layer showed high expression of VipR2 (SI table 1 and 2 and SI-Figure 2 and 3)."

We mention here that for the DISH experiment we used the channel 1 for either Vipr1 or Vipr2 probe(s) (weak blueish green punctate labelling which was very sensitive to dryness of the sample and more prone to fade with time) and VGAT mRNA probe in channel 1 (strong red labelling) so the blue signal can only be seen at high magnification. The photo-galleries presented in the Figure 3—figure supplement 1 and Figure 3—figure supplement 2, contained reactively high-resolution pictures but with high files sizes which may not be allowed to attach to the eventual publication. Hence, we plan to offer to provide the original files to interested readers.

Part 2: I think this part is the critical one in this manuscript. Starting from section 4, it uses the part 1 of the manuscript to review the literature and build a neuronal circuit with PACAP/PAC1 that makes for behavioural processes. It is literally a review inside the Results section. The schematic figures are interesting but also quite speculative regarding brain signalling since the authors did not performed any experiment to investigate the pathway of PACAP and the literature is scarce. Moreover, the role of Vip receptors were completely neglected here.

The main aim of the study was to place PACAP/PAC1 signalling within the context of the glutamate/GABA co-transmission in two scenarios, anatomical/embryological and basic sensorimotor circuits, to reveal its conspicuous role. We have added a short paragraph at the end of the Introduction aiming to better explain this design: " To address these issues, we conducted a systematic analysis placing the PACAP>PAC1 signalling into anatomical and basic sensorimotor circuit contexts. We first describe in detail the overall topographical organization of expression of PACAP mRNA and its predominant receptor PAC1 mRNA, and their co-expression with the small-molecule transmitters, glutamate and GABA, using VGLUT1, VGLUT2, and VGAT mRNAs, in mouse brain. We then examined the distribution of PACAP/PAC1 hubs within well-established sensory input-to-motor output pathways passing through the cognitive centers, within the context of glutamate/GABA neurotransmission. This systematic analysis has revealed several possible PACAP-dependent networks involved in sensory integration allowing environmental cues to guide motor output."

The so called "part 2" is a key part of this work because it provides a link between the anatomical findings and possible roles from system biology perspective and also toward identification of physiological factors, which could further up- or down-regulate the strength of this PACAP/PAC1 signalling pathway.

We also would like to state that the circuits we presented in this manuscript are all well-established basic circuits that most can be found in textbooks of physiology and classical literature, that it was not our aim to make any additions and to discuss the connectivity of any circuit components. They only serve the purpose for the present study as a functional Atlas / background, in resonance with the Allen Mouse Brain Atlas or Paxinos Mouse Atlas we used in the anatomical part, to project the PACAP-PAC1 hubs, to allow the functional significance of this peptide together with its receptor(s) and molecular signatures and anatomical locations to emerge at a systems level.